

# P-CSI v1.0, an accelerated barotropic solver for the high-resolution ocean model component in the Community Earth System Model v2.0

Xiaomeng Huang[1,2], Qiang Tang[1], Yuheng Tseng[3], Yong Hu[1], Allison H. Baker[3], Frank O. Bryan[3], John Dennis[3], Haohuan Fu[1], and Guangwen Yang[1]

[1]Ministry of Education Key Laboratory for Earth System Modeling, and Center for Earth System Science, Tsinghua University, Beijing, 100084, China
[2]Laboratory for Regional Oceanography and Numerical Modeling, Qingdao National Laboratory for Marine Science and Technology, Qingdao, 266237, China
[3]The National Center for Atmospheric Research, Boulder, CO, USA

*Correspondence to:* Xiaomeng Huang (hxm@tsinghua.edu.cn), Yuheng Tseng(ytseng@ucar.edu)

**Abstract.** In the Community Earth System Model (CESM), the ocean model is computationally expensive for high-resolution grids and is often the least scalable component for high-resolution production experiments. The major bottleneck is that the barotropic solver scales poorly at high core counts. We design a new barotropic solver to accelerate the high-resolution ocean simulation. The

novel solver adopts a Chebyshev-type iterative method to reduce the global communication cost in conjunction with an effective block preconditioner to further reduce the iterations. The algorithm and its computational complexity are theoretically analyzed and compared with other existing methods. We confirm the significant reduction of the global communication time with a competitive convergence rate using a series of idealized tests. Experimental results obtained with the CESM 0.1° global

ocean model show that the proposed approach results in a factor of 1.7 speed-up over the original method with no loss of accuracy, achieving 10.5 simulated years per wall-clock day on 16,875 cores.

## 1 Introduction

Recent progress in high-resolution global climate models has demonstrated that models with finer resolution can better represent important climate processes to facilitate climate prediction. Signifi-

cant improvements can be achieved in the high-resolution global simulations of Tropical Instability Waves (Roberts et al., 2009), El Niño Southern Oscillation (ENSO) (Shaffrey et al., 2009), the Gulf Stream separation (Chassignet and Marshall, 2008; Kuwano-Yoshida et al., 2010), the global water cycle (Demory et al., 2014), and other aspects of the mean climate and variability. Specifically, Gent et al. (2010) and Wehner et al. (2014) showed that increasing the atmosphere models' resolution

results in a better mean climate, more accurate depiction of the tropical storm formation, and more realistic events of extreme daily precipitation. Bryan et al. (2010) and Graham (2014) also suggested



that increasing the ocean models' resolution to the eddy resolving level helps capture the positive correlation between sea surface temperature and surface wind stress and improve the asymmetry of the ENSO cycle in the simulation.

In the High-Resolution Model Intercomparison Project (HighResMIP) for the Coupled Model Intercomparison Project phase 6 (CMIP6), global model resolutions of 25 km or finer at mid-latitudes are proposed to implement the Tier-1 and Tier-2 experiments (Eyring et al., 2015). Because all CMIP6 climate models are required to run for hundreds of years, tremendous computing resources are needed for high-resolution production simulations. To run high-resolution climate models prac-

tically, additional algorithm optimization is required to efficiently utilize the large-scale computing resources.

This work improves the barotropic solver performance in the ocean model component (Parallel Ocean Model, POP) of the National Center for Atmospheric Research (NCAR)'s fully coupled climate model: the Community Earth System Model (CESM). The POP solves the three-dimensional

primitive equations with hydrostatic and Boussinesq approximations and splits the time integration into two parts: the baroclinic and barotropic modes (Smith et al., 2010). The baroclinic mode describes the three-dimensional dynamic and thermodynamic processes, and the barotropic mode solves the vertically integrated momentum and continuity equations in two dimensions.

The barotropic solver is the major bottleneck in the POP within the high-resolution CESM because

it dominates the total execution time on a large number of cores (Jones et al., 2005). This results from the implicit calculation of the free-surface height in the barotropic solver, which scales poorly at the high core counts due to an evident global communication bottleneck inherent with the algorithm. The implicit solver allows a large time step to efficiently compute the fast gravity wave mode but requires a large elliptic system of equations to be solved. The conjugate gradient method (CG) and its variants

are popular choices in the implicit free-surface ocean solvers, such as MITgcm (Adcroft et al., 2014), FVCOM (Lai et al., 2010), MOM3 (Pacanowsky and Griffies, 1999), and OPA (Madec et al., 1997). However, the standard CG method has heavy global communication overhead in the existing POP implementation (Worley et al., 2011). The latest Chronopoulos-Gear (ChronGear) (D'Azevedo et al., 1999) variant of the CG algorithm is currently used in the POP to reduce the number of global

reductions. A nice overview of reducing global communication costs for CG method can be found in the work of Ghysels and Vanroose (2014). Recent efforts to improve the performance of CG method include a variant that overlaps the global reduction with the matrix-vector computation via a pipelined approach (Ghysels and Vanroose, 2014). However, the improvement is still limited when using a very large number of cores because of the remaining global reduction operations.

Another way to improve the CG method is preconditioning, which has been shown to effectively reduce the number of iterations. The current ChronGear solver in the POP has benefited by using a simple diagonal preconditioner (Pini and Gambolati, 1990; Reddy and Kumar, 2013). Some parallelizable methods such as polynomial, approximate-inverse, multigrid, and block preconditioning



have drawn much attention recently. High-order polynomial preconditioning can reduce iterations

as effectively as incomplete LU factorization in sequential simulations (Benzi, 2002). However, the computational overhead for the polynomial preconditioner typically offsets its superiority to the simple diagonal preconditioner (Meyer et al., 1989; Smith et al., 1992). The approximate-inverse preconditioner, although highly parallelizable, requires a linear system that is several times larger than the original system to be solved (Smith et al., 1992; Bergamaschi et al., 2007), which makes it

less attractive for the POP.

The multigrid method is another well-known scalable and efficient approach to solve the elliptic systems and is commonly used as a preconditioner in the sequential models. Recent works indicated that the geometric multigrid is promising in atmosphere and ocean modeling (Müller and Scheichl, 2014; Matsumura and Hasumi, 2008; Kanarska et al., 2007). However, the geometric multigrid in

global ocean models does not always scale ideally because of the presence of complex topography, non-uniform or anisotropic grids (Fulton et al., 1986; Stüben, 2001; Tseng and Ferziger, 2003; Matsumura and Hasumi, 2008). The current POP, which employs general orthogonal girds to avoid the pole singularity, is a typical example. This leads to an elliptic system with variable coefficients defined on an irregular domain with non-uniform grids. The algebraic multigrid (AMG) is an alter-

native to the geometric multigrid to handle complex topography. However, the AMG setup in the parallel environment is more expensive than the iterative solver in climate modelling, which makes it unfavorable as a preconditioner (Müller and Scheichl, 2014).

Block preconditioning has been shown to be an effective parallel preconditioner (Concus et al., 1985; White and Borja, 2011) and is appealing for the POP because it uses the block structure of

the coefficient matrix that arises from the discretization of the elliptic equations. This advantage can further improve solver parallel performance. Some other algorithmic approaches also attempt to improve the parallel performance of ocean models. For example, a load-balancing algorithm based on the space-filling curve was proposed that not only eliminates land blocks but also reduces the communication overhead due to the reduced number of processes (Dennis, 2007; Dennis and Tufo,

2008). Beare and Stevens (1997) also proposed increasing the number of extra halos and communication overlaps in the parallel ocean general circulation. Although these approaches improve the performance of ocean models, the global communication bottleneck still exists.

To improve the scalability of the POP at the high core counts, we abandon the CG-type approach and design a new barotropic solver that does not include global communication in iteration steps.

The new barotropic solver, named P-CSI, uses a Classical Stiefel Iteration (CSI) method (proposed originally in Hu et al., 2015) with an efficient block preconditioner based on the Error Vector Propagation (EVP) method (Roache, 1995). The P-CSI solver is now the default ocean barotropic solver for the upcoming CESM 2.0 release. This paper extends our conference paper (Hu et al., 2015) presented at the 27th International Conference for High Performance Computing, Networking, Storage



and Analysis (SC2015) to emphasize the theoretical analysis of the computational complexity, the convergence of P-CSI and the high-resolution POP results.

The remainder of this paper is organized as follows. Section 2 reviews the existing barotropic solver in the POP. Sections 3 details the design of the P-CSI solver, followed by an analysis of the computational complexity and convergence rate of P-CSI in Section 4. Section 5 further compares the high-resolution performance of the existing solvers and the P-CSI solvers. Finally, conclusions are given in Section 6.

## 2 Barotropic solver background

We briefly describe the governing equations to formally derive the new P-CSI solver in the POP. The primitive momentum and continuity equations are expressed as:

$$\frac{\partial}{\partial t}\mathbf{u} + \mathcal{L}(\mathbf{u}) + f \times \mathbf{u} = -\frac{1}{\rho_0}\nabla p + F_H(\mathbf{u}) + F_V(\mathbf{u}), \tag{1}$$

$$\mathcal{L}(1) = 0, \tag{2}$$

where $\mathcal{L}(\alpha) = \frac{\partial}{\partial x}(u\alpha) + \frac{\partial}{\partial y}(v\alpha) + \frac{\partial}{\partial z}(w\alpha)$, which is equivalent to the divergence operator when $\alpha = 1$; $x, y,$ and $z$ are the horizontal and vertical coordinates; $\mathbf{u} = [u, v]^T$ is the horizontal velocity; $w$ is the vertical velocity; $f$ is the Coriolis parameter; $p$ and $\rho_0$ represent the pressure and the water density, respectively; and $F_H$ and $F_V$ are the horizontal and vertical dissipative terms, respectively (Smith et al., 2010). In particular, we emphasize the two-dimensional barotropic mode in the time-splitting scheme, where the P-CSI is implemented .

### 2.1 Barotropic mode

The governing equations for the barotropic mode can be obtained by vertically integrating Eq. (1) and Eq. (2) from the ocean bottom topography to the sea surface:

$$\frac{\partial \mathbf{U}}{\partial t} = -g\nabla\eta + F, \tag{3}$$

$$\frac{\partial \eta}{\partial t} = -\nabla \cdot H\mathbf{U} + q_w, \tag{4}$$

where $\mathbf{U} = \frac{1}{H+\eta}\int_{-H}^{\eta} dz\mathbf{u}(z) \approx \frac{1}{H}\int_{-H}^{0} dz\mathbf{u}(z)$ is the vertically integrated barotropic velocity, $g$ is the acceleration due to gravity, $\eta$ is the sea surface height (defined as $p_s/\rho_0 g$, where $p_s$ is the surface pressure associated with undulations of the free surface), $H$ is the depth of the ocean bottom, $q_w$ is the freshwater flux per unit area, and $F$ is the vertical integral of all other terms except the time-tendency and surface pressure gradient in the momentum Eq. (1). To simplify the solution procedure, the barotropic continuity Eq. (4) is linearized by dropping the term involving $\nabla\eta$ in the boundary condition (Smith et al., 2010).




Equation (3) and Eq. (4) are then discretized in time using an implicit scheme as follows:

$$\frac{\mathbf{U}^{n+1} - \mathbf{U}^n}{\tau} = -g\nabla\eta^{n+1} + F, \tag{5}$$

$$\frac{\eta^{n+1} - \eta^n}{\tau} = -\nabla \cdot H\mathbf{U}^{n+1} + q_w, \tag{6}$$

where $\tau$ is the time step associated with the time advance scheme. By replacing the barotropic velocity in Eq. (6) with the barotropic velocity at the next time step in Eq. (5), an elliptic system of

sea surface height $\eta$ is obtained

$$[-\nabla \cdot H\nabla + \frac{1}{g\tau^2}]\eta^{n+1} = -\nabla \cdot H[\frac{\mathbf{U}^n}{g\tau} + \frac{F}{g}] + \frac{\eta^n}{g\tau^2} + \frac{q_w}{g\tau}. \tag{7}$$

For simplicity, we can rewrite the elliptic Eq. (7) as

$$[-\nabla \cdot H\nabla + \frac{1}{g\tau^2}]\eta^{n+1} = \psi(\eta^n, \tau), \tag{8}$$

where $\psi$ represents a function of the current state of $\eta$.

Spatially, the POP utilizes the Arakawa B-grid on the horizontal grid (Smith et al., 2010) with the following nine-point stencils to discretize Eq. (8) as follows (see Fig. 1):

$$\nabla \cdot H\nabla\eta = \frac{1}{\Delta y}\delta_x\overline{[\Delta y H \delta_x \overline{\eta}^y]}^y + \frac{1}{\Delta x}\delta_y\overline{[\Delta x H \delta_y \overline{\eta}^x]}^x, \tag{9}$$

where $\delta_\xi$ ($\xi \in \{x,y\}$) are finite differences and $\Delta_\xi$ ($\xi \in \{x,y\}$) are the associated grid lengths. The finite difference $\delta_\xi(\psi)$ and average $\overline{\psi}^\xi$ notations are defined, respectively, as follows:

$$\delta_\xi\psi = [\psi(\xi + \Delta_\xi/2) - \psi(\xi - \Delta_\xi/2)]/\Delta_\xi, \tag{10}$$

$$\overline{\psi}^\xi = [\psi(\xi + \Delta_\xi/2) + \psi(\xi - \Delta_\xi/2)]/2. \tag{11}$$

Because the POP uses general orthogonal girds, the coefficient matrix varies in space. To demonstrate the properties of the sparse matrix used in the POP, we can simplify Eq. (9) using a special case with uniform grids and constant ocean depth $H$ as follows:

$$[\nabla \cdot H\nabla\eta]_{i,j} = -\frac{H}{S_{i,j}}(A_{i,j}^O\eta_{i,j} + A_{i,j}^{NW}\eta_{i-1,j+1} + A_{i,j}^N\eta_{i,j+1} + A_{i,j}^{NE}\eta_{i+1,j+1} + A_{i,j}^W\eta_{i-1,j}$$

$$+ A_{i,j}^E\eta_{i+1,j} + A_{i,j}^{SW}\eta_{i-1,j-1} + A_{i,j}^S\eta_{i,j-1} + A_{i,j}^{SE}\eta_{i+1,j-1}), \tag{12}$$

where $S_{i,j} = \Delta x \Delta y$, $A_{i,j}^\chi(\chi \in \mathcal{Q} = \{O, NW, NE, SW, SE, W, E, N, S\})$ are coefficients between grid point $(i,j)$ and its neighbors using the nine-point stencil discretization (9), as determined by $\Delta x, \Delta y, \tau$ and $H$:

$$\alpha = \frac{\Delta y}{\Delta x}, \quad \beta = 1/\alpha,$$

$$A_{i,j}^{NW} = A_{i,j}^{NE} = A_{i,j}^{SW} = A_{i,j}^{SE} = -(\alpha + \beta)/4,$$

$$A_{i,j}^W = A_{i,j}^E = (\beta - \alpha)/2, \tag{13}$$

$$A_{i,j}^N = A_{i,j}^S = (\alpha - \beta)/2,$$

$$A_{i,j}^O = \alpha + \beta.$$



The full discretization of Eq. (8) for any given grid point $(i,j)$ can then be written as

$$(A_{i,j}^O + \phi)\eta_{i,j} + A_{i,j}^{NW}\eta_{i-1,j+1} + A_{i,j}^N\eta_{i,j+1} + A_{i,j}^{NE}\eta_{i+1,j+1} + A_{i,j}^W\eta_{i-1,j}$$
$$+ A_{i,j}^E\eta_{i+1,j} + A_{i,j}^{SW}\eta_{i-1,j-1} + A_{i,j}^S\eta_{i,j-1} + A_{i,j}^{SE}\eta_{i+1,j-1} = \frac{S_{i,j}}{H}\psi_{i,j}, \tag{14}$$

where $\phi = \frac{S_{i,j}}{g\tau^2 H}$ is a factor of the time step.

Therefore, the elliptic Eq. (7) leads to a linear system of $\eta$, i.e., $Ax = b$, where $A$ is a block tridiagonal matrix composed of coefficients $A_{i,j}^\chi (\chi \in \mathcal{Q})$. The simplified equation set of (13) and (14) show that $A$ is mainly determined by the horizontal grid sizes, ocean depth and time step. These impacts will be further discussed in Section 4.1. Note that Eq. (14) also indicates that the sparse pattern of A comes directly from the nine nonzero elements in each row (Fig. 2).

**2.2   Barotropic solvers**

The barotropic solver in the original POP uses the PCG method with a diagonal preconditioner $M = \Lambda(A)$ because of its efficiency in small-scale parallelism (Dukowicz and Smith, 1994) (see Appendix A1 for the details). To mitigate the global communication bottleneck, ChronGear, a variant of the CG method proposed by D'Azevedo et al. (1999), was later introduced as the default solver
in the POP. It combines the two separated global communications of a single scalar into a single global communication (see Appendix A2). By this strategic rearrangement, the ChronGear method achieves a one-third latency reduction in the POP. However, the scaling bottleneck still exists in the high-resolution POP using this solver, particularly with a large number of cores (Fig. 3).

To accurately profile the parallel cost of the barotropic solvers, we clearly separate the timing for
computation, boundary communication and global reduction. Operations such as scaler computations and vector scalings are categorized as pure computations, which are relatively cheap due to the independent operations on each process. The extra boundary communication is required for each process to update the boundary values from its neighbors (Fig. 1) after the matrix-vector multiplication. This boundary communication usually costs more than the computation when a large number
of cores is used (due to a decreasing problem size per core). The global reduction, which is needed by the inner products of vectors, is even more costly (Hu et al., 2013). Worley et al. (2011) and Dennis et al. (2012) specifically indicated that the global reduction in the POP's barotropic solver is the main scaling bottleneck for the high-resolution ocean simulation.

Figure 3 confirms that the percentage of execution time for the barotropic mode in 0.1° POP
indeed increases with increasing number of processor cores on Yellowstone. When 470 cores are used, the execution time of the barotropic (baroclinic) solver is approximately 5% (90%) of the total execution time (excludes initialization and I/O). However, when several thousand cores are used, the percentage of time spent in the baroclinic mode decreases, associated with the increasing percentage of time in the barotropic solver. With more than sixteen thousand cores, the percentage of the total
execution time due to the barotropic solver is nearly 50%.



## 3  Design of the P-CSI solver

The CG-type solver converges rapidly in the sequential computation (Golub and Van Loan, 2012). However, the bottleneck of global communication embedded in ChronGear still limits the large-scale parallel performance. Here, we design a new solver with the goal of reducing global communication so that the speed-up can be as close to unity as possible when a significant number of cores is used.

### 3.1  Classical Stiefel Iteration method

The CSI is a special type of Chebyshev iterative method (Stiefel, 1958). Saad et al. (1985) proposed a generalization of CSI on linearly connected processors and claimed that this approach outperforms the CG method when the eigenvalues are known. This method was revisited by Gutknecht and Röllin (2002) and shown to be ideal for massively parallel computers. In the procedure of preconditioned CSI (P-CSI, details are provided in Appendix A3), the iteration parameters, which control the searching directions in the iteration step, are derived from a stretched Chebyshev function of two extreme eigenvalues (Stiefel, 1958). We demonstrate in Section 4.2 that the stretched Chebyshev function in P-CSI provides a series of preset parameters for iteration directions. As a result, P-CSI requires no inner product operation, thus potentially avoiding the bottleneck of global reduction (see the workflow of ChronGear and P-CSI in Fig. 4). This makes the P-CSI more scalable than ChronGear on massively parallel architectures. However, it requires *a priori* knowledge about the spectrum of coefficient matrix $A$ (Gutknecht and Röllin, 2002). It is well known that obtaining the eigenvalues of a linear system of equations is equivalent to solving it. Fortunately, the coefficient matrix $A$ and its preconditioned form in the POP are both positive definite real symmetric matrices. Approximate estimation of the largest and smallest eigenvalues, $\mu$ and $\nu$, respectively, of the preconditioned coefficient matrix is sufficient to ensure the convergence of P-CSI.

To efficiently estimate the extreme eigenvalues of the preconditioned matrix $M^{-1}A$ (where $M$ is the preconditioner), we adopt the Lanczos method  (Paige, 1980) (see the algorithm in Appendix B). Initial tests indicate that only a small number of Lanczos steps is necessary to reasonably estimate the extreme eigenvalues of $M^{-1}A$ that results in the near-optimal P-CSI convergence (Hu et al., 2015). Therefore, the computational overhead of the eigenvalue estimation is very small in our algorithm.

### 3.2  A block EVP preconditioner

Block preconditioning is quite promising in POP because the parallel domain-decomposition is ideal for the block structure. A block preconditioning based on the EVP method is proposed and detailed in Hu et al. (2015) to improve the parallel performance of the barotropic solver in the POP. To the best of our knowledge, the EVP and its variants are among the least costly algorithms for solving elliptic equations in serial computation (Roache, 1995), which have also been used in several different Ocean models (Dietrich et al., 1987; Sheng et al., 1998; Young et al., 2012). The parallel EVP solver was



also implemented by Tseng and Chien (2011). The standard EVP is actually a direct solver, which requires two solution steps: preprocessing and solving. In the preprocessing stage, the influence coefficient matrix and its inverse are computed, involving a computational complexity of $\mathcal{C}_{pre} = (2n-5)*9n^2 + (2n-5)^3 = \mathcal{O}(26n^3)$, which is intensive but computed only once at the beginning. The solving stage is computed at every time step and requires only $\mathcal{C}_{evp} = 2*9n^2 + (2n-5)^2 =$

$\mathcal{O}(22n^2)$ (Hu et al., 2015), which is a much lower computational cost than other direct solvers such as LU.

The EVP method is efficient for solving elliptic equations. However, a major drawback of the standard EVP is that, without applying additional modifications, it cannot be used for a large domain due to its global error propagation, which will cause arithmetic overflow in the marching process

(Roache, 1995). The fact that the EVP is not well-suited for large domains is not an issue for large-scale parallel computing, where a larger number of processors typically results in smaller domains. Thus, the serial disadvantage becomes an advantage in parallel computing, making the EVP ideal for parallel block preconditioning on a large number of cores. Although the EVP preconditioning may increase the required computation for each iteration, the barotropic solver can greatly benefit from

the resulting reduction in iterations, particularly at very large numbers of cores when communication costs dominate the total costs (Hu et al., 2015). We will further illustrate this advantage in Section 5.2.3.

## 4 Algorithm analysis and comparison

The extreme eigenvalues of the coefficient matrix are critical to determine the convergence of the

iterative solvers (such as P-CSI, PCG and ChronGear). Here, the characteristics of P-CSI are investigated in terms of the associated eigenvalues and their connection with the convergence rate. The computational complexity is also addressed.

### 4.1 Spectrum and condition number

Because the coefficient matrix $A$ in POP is symmetric and positive-definite (Smith et al., 2010),

its eigenvalues are positive real numbers (Stewart, 1976). We assume that the spectrum (Golub and Van Loan, 2012) of A is $\mathcal{S} = \{\lambda_1, \lambda_2, \cdots, \lambda_N\}$, where $\lambda_{min} = \lambda_1 \leq \lambda_i \leq \lambda_{\mathcal{N}} = \lambda_{max}$ ( $1 < i < \mathcal{N}$, $\mathcal{N}$ is the size of $A$) are the eigenvalues of $A$. The condition number, defined as $\kappa = \lambda_{max}/\lambda_{min}$, is determined by the spectrum radius. Using the Gershgorin circle theorem (Bell, 1965), we know that for any $\lambda \in \mathcal{S}$, there exists a pair of $(i,j)$ satisfying

$$|\lambda - (A_{i,j}^O + \phi)| \leq \sum_{\chi \in \mathcal{Q} - \{O\}} |A_{i,j}^\chi|, \tag{15}$$



where $\phi = \frac{S}{g\tau^2 H}$ is defined in Section 2.1. With the definition of the coefficients in (13), we obtain

$$\lambda_{max} \leq \max(5\alpha - \frac{1}{\alpha}, \frac{5}{\alpha} - \alpha) + \phi,$$
$$\lambda_{min} \geq 2\min(\alpha - \frac{1}{\alpha}, \frac{1}{\alpha} - \alpha) + \phi. \tag{16}$$

To quantitatively evaluate the impacts of the condition number, we set up a series of idealized test cases to solve Eq. (8) in which the coefficient matrices are derived from Eq. (13) and (14) on an idealized cylinder with an earth-size perimeter, which is $2\pi R$ (radius $R$ is 6372 km), and a height of $\pi R$. A uniform grid with a size of $N \times M$ is used, where the grid size along the perimeter and height are $\Delta x = 2\pi R/N$ and $\Delta y = \pi R/M$, respectively. The depth $H$ is $4km$.

The inequalities (16) suggest that the lower bound of eigenvalues is mostly determined by $\phi$. This indicates that for a given ocean configuration and grid size, the lower bound of the eigenvalues will decrease with increasing time step, resulting in a larger condition number. Figure 5 shows the condition number versus the time step size when the total number of grid points is a constant $\mathcal{N} = N \times M = 2048$. Three different grid decompositions ($32 \times 64$, $64 \times 32$ and $128 \times 16$) are shown to reflect the influence of different grid aspect ratios. When the size of the time step is sufficiently small (smaller than $10^6$s), $\phi$ in all cases becomes very large and dominates both $\lambda_{max}$ and $\lambda_{min}$. As a result, the condition number is close to 1. However, when the size of the time step is large enough (larger than $10^8$s), the condition number is highly determined by the grid aspect ratio $\alpha$ because of the small $\phi$.

Consistent with the theoretical bounds of the extreme eigenvalues in Eq. (16), the condition number in Fig. 5 is smallest when the grid aspect ratio is close to unity (i.e., the decomposition of $64 \times 32$). When the aspect ratio of the horizontal grid cell is close to unity, the upper (lower) bound of the largest (smallest) eigenvalue decreases (increases), leading to a reduced spectrum radius ($[\lambda_{min}, \lambda_{max}]$). This implies that the condition number is also reduced at the same time. When the aspect ratio equals to unity (i.e., $\alpha = \frac{\Delta y}{\Delta x} = 1$), we obtain $\lambda_{max} \leq 4 + \phi$ and $\lambda_{min} \geq \phi$. Figure 6 shows the condition number versus the aspect ratio with fixed grid size $\mathcal{N} = 2048$. Three different time step sizes are tested: $1.0 \times 10^5 s$, $5.0 \times 10^5 s$ and $10.0 \times 10^5 s$. Under this configuration, it is clear to see that the condition number reaches its minimum when the aspect ratio is unity, regardless of the time step size.

When the time step is sufficiently large, the foregoing analysis indicates that the spectrum radius is confined in $(\phi, 4 + \phi)$ if the aspect ratio is 1 regardless of grid sizes. However, the condition number may vary greatly because when the grid size $\mathcal{N}$ increases, the largest eigenvalue remains close to 4, whereas the smallest eigenvalue becomes closer to $\phi$. The previous discussion implies that the condition number is significantly affected when the aspect ratio is far from unity. To focus on the impact of the number of grid points, we choose a constant aspect ratio. Because different time step sizes may play an important role when the grid size increases, we assumed that the time step must satisfy the Courant-Friedrichs-Lewy (CFL) condition (Courant et al., 1967), that is, $\tau = \frac{\Delta x}{v}$,





where $v$ is the supported barotropic velocity. Three different configurations of the time step based on $v = 2m/s$, $v = 20m/s$ and $v = 200m/s$ are chosen. Figure 7 shows that the condition number increases monotonically with increasing grid size. It also shows that the time step (specified by a different propagating speed) has a large impact on the condition number.

### 4.2 Convergence rate

The convergence rate of any elliptic solver relies heavily on the condition number of the preconditioned coefficient matrix $A'$. Both PCG and ChronGear have the same theoretical convergence rate because of the same numerical algorithm but different implementations (D'Azevedo et al., 1999). Their relative residual in the k-th iteration has an upper bound as follows (Liesen and Tichý, 2004):

$$\frac{||\mathbf{x}_k - \mathbf{x}^*||_{A'}}{||\mathbf{x}_0 - \mathbf{x}^*||_{A'}} \leq \min_{p \in \mathcal{P}_k, p(0)=1} \max_{\lambda \in \mathcal{S}} |p(\lambda)|, \tag{17}$$

where $\mathbf{x}_k$ is the solution vector after the k-th iteration, $\mathbf{x}^*$ is the solution of the linear equation (i.e., $\mathbf{x}^* = A^{-1}b$), $\lambda$ represents an eigenvalue of $A'$, and $\mathcal{P}_k$ is the vector space of polynomials with real coefficients and degree less than or equal to $k$. Applying the Chebyshev polynomials of the first kind to estimate this min-max approximation, we obtain

$$||\mathbf{x}_k - \mathbf{x}^*||_{A'} \leq 2(\frac{\sqrt{\kappa}-1}{\sqrt{\kappa}+1})^k ||\mathbf{x}_0 - \mathbf{x}^*||_{A'}, \tag{18}$$

where $\kappa = \kappa_2(A') = \frac{\lambda'_{max}}{\lambda'_{min}}$ is the condition number of the matrix $A'$ with respect to the $l_2$-norm. Equation (18) indicates that the theoretical bound of the convergence rate of PCG decreases with increasing condition number. PCG converges faster for a well-conditioned matrix (e.g., a matrix with a small condition number) than an ill-conditioned matrix.

We now show that the P-CSI has the same order of convergence rate as PCG and ChronGear with the additional advantage of fewer global reductions in parallel computing. With the estimated smallest and largest extreme eigenvalues of coefficient matrix $\nu$ and $\mu$, the residual for the P-CSI algorithm satisfies

$$\mathbf{r}_k = P_k(A')\mathbf{r}_0, \tag{19}$$

where $P_k(\zeta) = \frac{\tau_k(\beta - \alpha\zeta)}{\tau_k(\beta)}$ for $\zeta \in [\nu, \mu]$ (Stiefel, 1958). $\tau_k(\xi)$ is a Chebyshev polynomial expressed as

$$\tau_k(\xi) = \frac{1}{2}[(\xi + \sqrt{\xi^2 - 1})^k + (\xi + \sqrt{\xi^2 - 1})^{-k}]. \tag{20}$$

When $\xi \in [-1, 1]$, the Chebyshev polynomial has an equivalent form

$$\tau_k(\xi) = cos(k \cos^{-1} \xi), \tag{21}$$

which clearly shows that $|\tau_k(\xi)| \leq 1$ when $|\xi| \leq 1$. $P_k(\zeta)$ is the polynomial satisfying that

$$P_k = \min_{p \in \mathcal{P}_k, p(0)=1} \max_{\zeta \in [\nu, \mu]} |p(\zeta)|. \tag{22}$$





Assume that $A' = Q^T \Lambda Q$, where $\Lambda$ is a diagonal matrix having the eigenvalues of $A'$ on the diagonal, and $Q$ is a real orthogonal matrix with the columns that are eigenvectors of $A'$. We then have

$$P_k(A') = Q^T P_k(\Lambda) Q = Q^T \begin{bmatrix} P_k(\lambda_1) & & & \\ & P_k(\lambda_2) & & \\ & & \ddots & \\ & & & P_k(\lambda_N) \end{bmatrix} Q. \tag{23}$$

Assuming that $\nu$ and $\mu$ satisfy $0 < \nu \leq \lambda_i \leq \mu$ ($i = 1, 2, \cdots, \mathcal{N}$), then Eq. (21) indicates that $|\beta - \alpha \lambda_i| \leq 1$ and $|P_k(\lambda_i)| = \frac{\tau_k(\beta - \alpha \lambda_i)}{\tau_k(\beta)} \leq \tau_k^{-1}(\beta)$. Equations (19) and (23) indicate that

$$\frac{||\mathbf{r}_k||_2}{||\mathbf{r}_0||_2} \leq \tau_k^{-1}(\beta) = \frac{2(\beta + \sqrt{\beta^2 - 1})^k}{1 + (\beta + \sqrt{\beta^2 - 1})^{2k}} \leq 2(\frac{\sqrt{\kappa'} - 1}{\sqrt{\kappa'} + 1})^k, \tag{24}$$

where $\kappa' = \frac{\mu}{\nu}$. Equation (24) shows that the P-CSI has the same theoretical upper bound of convergence rate as PCG and ChronGear when the estimation of eigenvalues is appropriate (e.g., $\kappa' = \kappa$)
.

The foregoing analysis applies to cases in which a nontrivial preconditioning is used. Assume that the preconditioned coefficient matrix $A' = M^{-1}A$. It is worth mentioning that the preconditioned matrix in the PCG, ChronGear and P-CSI algorithms is actually $M^{-1/2}A(M^{-1/2})^T$, which is symmetric and has the same set of eigenvalues as $M^{-1}A$ (Shewchuk, 1994). Thus, the condition number of the preconditioned matrix is $\kappa = \kappa_2(M^{-1/2}A(M^{-1/2})^T)$, which is usually smaller than the condition number of $A$. The closer $M$ is to $A$, the smaller the condition number of $M^{-1}A$ is. When $M$ is the same as $A$, then $\kappa_2(M^{-1}A) = 1$.

Because the convergence rate of P-CSI is on the same order as PCG and ChronGear, the performance between P-CSI and the CG-type solvers should be comparable when a small number of cores is used. When a large number of cores is used for the high-resolution ocean model, P-CSI should be significantly faster than PCG or ChronGear per iteration due to the bottleneck in the CG-type method. This is shown in the following analysis of computational complexity.

### 4.3 Computational complexity

To analyze the computational complexity of P-CSI and compare it with ChronGear, we assume that $p$ is the number of processes and $\mathcal{N}$ is the number of grid points following the same definition as in Hu et al. (2015). Both ChronGear and P-CSI solver time can then be divided into three major components: computation, boundary updating, and global communication. The complexity of computation varies among different solvers and preconditioners. The boundary communication complexity is $\mathcal{T}_b = \mathcal{O}(4\varpi + 8\sqrt{\frac{\mathcal{N}}{p}}\vartheta)$, where $\varpi$ is the ratio of point-to-point communication latency per message to the time of one floating-point operation and $\vartheta$ is the ratio of the transfer time per byte (inverse of bandwidth) to the time of one floating-point operation. All boundary update times show



a similarly decreasing trend with increasing number of processes but have a lower bound $4\varpi$. The global communication exists only in the ChronGear solver and contains one global reduction per iteration, resulting from the MPI_Allreduce and a masking operation to exclude land points. The cost of the masking operation decreases with increasing processes $p$, whereas the cost of MPI_Allreduce monotonically increases, so the global reduction complexity satisfies $\mathcal{T}_g = \mathcal{O}(2\frac{\mathcal{N}}{p} + \log p\varpi)$.

The execution time of one diagonal preconditioned ChronGear solver step can then be expressed as:

$$\mathcal{T}_{cg} = \mathcal{K}_{cg}(\mathcal{T}_c + \mathcal{T}_b + \mathcal{T}_g) = \mathcal{O}(\mathcal{K}_{cg}(18\frac{\mathcal{N}}{p} + 8\sqrt{\frac{\mathcal{N}}{p}}\vartheta + (4 + \log p)\varpi), \tag{25}$$

where $\mathcal{K}_{cg}$ is the number of iterations, which does not change with the number of processes (Hu et al., 2015). The complexity of P-CSI with a diagonal preconditioner is

$$\mathcal{T}_{pcsi} = \mathcal{O}(\mathcal{K}_{pcsi}(12\frac{\mathcal{N}}{p} + 8\sqrt{\frac{\mathcal{N}}{p}}\vartheta + 4\varpi)), \tag{26}$$

where $\mathcal{K}_{pcsi}$ is the number of iterations.

Equation (25) indicates that the computation and boundary update time decreases with increasing number of processes. However, the time required for the global reduction increases with increasing numbers of processes. Therefore, we can expect the execution time of the ChronGear solver to increase when the number of processors exceeds a certain threshold. Our analysis shows that P-CSI has a lower computational complexity than ChronGear due to the lack of a $\log p$ term associated with global communications.

We further consider the computational complexity of preconditioning. The EVP preconditioning has $\mathcal{O}(22\frac{\mathcal{N}}{p})$. Thus, with the EVP preconditioning, the computational complexity of ChronGear and P-CSI becomes $\mathcal{O}(39\frac{\mathcal{N}}{p})$ and $\mathcal{O}(33\frac{\mathcal{N}}{p})$, respectively. As a result, the total complexities of ChronGear and P-CSI with EVP preconditioning are

$$\mathcal{T}_{cg-evp} = \mathcal{O}(\mathcal{K}_{cg-evp}(39\frac{\mathcal{N}}{p} + 8\sqrt{\frac{\mathcal{N}}{p}}\vartheta + (4 + \log p)\varpi), \tag{27}$$

$$\mathcal{T}_{pcsi-evp} = \mathcal{O}(\mathcal{K}_{pcsi-evp}(33\frac{\mathcal{N}}{p} + 8\sqrt{\frac{\mathcal{N}}{p}}\vartheta + 4\varpi)). \tag{28}$$

Although the computation time in each iteration doubles with the EVP preconditioning, the total time may still decrease if the number of iterations is reduced. Indeed, with EVP preconditioning, the iteration number $\mathcal{K}_{pcsi-evp}$ decreases by almost one-half (see Fig. 11). As a result, the total number of communications, which is the most time-consuming part on a large number of cores, decreases by approximately one-half.



## 5 Numerical experiments

To evaluate the new P-CSI solver, we first demonstrate its characteristics and compare it with PCG
(and thus ChronGear) using an idealized test case. The actual performance of P-CSI in CESM POP
is then evaluated and compared with the existing solvers using the $0.1°$ high-resolution simulation.

### 5.1 Condition number and convergence rate

To confirm the theoretical analysis of the convergence in Section 4.2, we created a series of ma-
trices with the idealized setting illustrated in Section 4.1. Instead of a cylindrical grid, we choose
a spherical grid with two polar continents (ocean latitude varies from $80°$S to $80°$N). A uniform
latitude-longitude grid is used, in which the grid size along the longitude varies with latitude coordi-
nate $\theta$, that is, $\Delta x = \pi R \cos\theta$. The time step size is set to $\tau = \frac{\Delta x}{v}$, where $v = 2m/s$ is the barotropic
velocity of the ocean water as used in Section 4.1. These cases differ in the number of grid points,
so the condition numbers vary. We compare the results using PCG and P-CSI solvers with no pre-
conditioning, diagonal preconditioning and EVP preconditioning, respectively. Here, the block size
in EVP preconditioning is set to be $5 \times 5$, and the convergence tolerance is $tol = 10^{-6}$. We note that
the theoretical convergence rates of ChronGear and PCG are identical, so the results here can apply
to the ChronGear at the same time.

As shown in Fig. 8, as the problem size increases, the coefficient matrix becomes more poorly
conditioned. All solvers must iterate more to obtain the same level of relative residual. For both
PCG and P-CSI, the convergence rate varies with different preconditioners. Given the same problem
size, the solvers without preconditioning have the most iterations, and those using the EVP precon-
ditioning require the fewest iterations. This confirms that, with the EVP preconditioning, the matrix
becomes better conditioned than the one without preconditioning or with diagonal preconditioning.
It also shows that with the same preconditioning, P-CSI has a slower convergence rate than PCG. It
is worth mentioning that the diagonal preconditioner improves the convergence only slightly in our
idealized cases because the grid is uniform and the ocean depth is constant in this configuration.

### 5.2 A practical application using the high-resolution CESM POP

#### 5.2.1 Experiment platform and configuration

We evaluate the performance of P-CSI in CESM1.2.0 on the Yellowstone supercomputer, located at
NCAR-Wyoming Supercomputing Center (NWSC) (Loft et al., 2015). Yellowstone uses Intel Xeon
E5-2670 (Sandy Bridge: 16 cores @ 2.6 GHz, hyperthreading enabled, 20 MB shared L3 cache)
and provides a total of 72,576 cores connected by a 13.6 GBps InfiniBand network. More than 50%
of Yellowstone's cycles are consumed by CESM. Therefore, the ability to accelerate the parallel
performance on Yellowstone is critical to support the CESM production simulations.





To emphasize the advantage of P-CSI, we use the finest $0.1°$ grid and 60 vertical levels POP with the "G_NORMAL_YEAR" configuration, which uses active ocean and sea ice components (i.e., the atmosphere and land components are replaced by pre-determined forcing data sets). The I/O optimization is another important issue for the high-resolution POP (Huang et al., 2014) but is not addressed here.

The choice of ocean block size and layout has a large impact on performance for the high-resolution POP because it directly affects the distribution of the workload among processors. To remove the influence of different block distribution on our results, we carefully specify block decompositions for each core with the same ratio. The time step is set to the default of 172.8 seconds. For a fair comparison among solvers, the convergence is checked every 10 iterations for all tests. The impacts of CSI and the EVP preconditioner are evaluated separately using several different numerical experiments.

### 5.2.2 Performance of CSI

The first experiment compares the parallel performance among the three solvers using the same diagonal preconditioners: PCG, ChronGear and P-CSI. Figure 9 compares the convergence rate (relative residual versus the number of iterations) among them. Although the order of computation in ChronGear differs slightly with that in PCG, their convergence rates are almost identical as expected. P-CSI converges slightly slower than PCG and ChronGear at the beginning and the final iteration steps, which is related to the unstable distribution of the coefficient matrix's eigenvalues. However, all of these solvers have very similar order of slopes, thus supporting the same upper bound of convergence rate discussed in Section 4.2.

Through a number of experiments, we set the Lanczos convergence tolerance $\epsilon$ to $0.15$ to obtain the balance between fast convergence rate and reasonable relative residual at the same time. Generally, we can estimate the largest and smallest eigenvalues in no more than 50 Lanczos steps. This causes P-CSI to result in near-optimal convergence.

Figure 10 further evaluates the timing for the different phases in the solver. It is clear that P-CSI outperforms ChronGear primarily because it only requires a few global reductions in convergence check. No obvious difference can be found for the boundary updates and the computation phases when using large core counts. The reduction in global communications will also significantly reduce the sensitivity of the ocean model component to operating system noise (Ferreira et al., 2008) by increasing the time interval between global synchronizations.

### 5.2.3 Performance of EVP preconditioner

The second experiment evaluates the performance of the EVP preconditioner used in the P-CSI solver by comparing the CSI solvers with no preconditioner, the diagonal preconditioner and the EVP preconditioner. Figure 11 shows that the preconditioners can effectively reduce the number





of iterations. The standard CSI without any preconditioner requires 350 iterations to achieve $10^{-15}$ relative residual. The iterations are significantly reduced to approximately 100 and 200 steps for EVP and diagonal preconditioners, respectively. This confirms that the preconditioned matrix $M^{-1}A$ indeed has a smaller condition number than the original matrix $A$ and effectively accelerates the
convergence without any consideration of parallelization.

As a result, the EVP preconditioner reduces not only the execution time of global reduction but also the execution time of boundary update owing to the reduced iterations (Fig. 12). All of these results are consistent with the theoretical analysis in Section 4.3. Note that the extra computation operations required by the EVP preconditioner have only a small impact on the overall performance
of the barotropic solver.

### 5.2.4 Parallel performance

The last experiment compares the simulated speeds of P-CSI and ChronGear on a variety of computing cores, ranging from 470 to 16,875 cores. When the timing refers to the barotropic mode calculation only, we find that the performance of the ChronGear solver begins to degrade after ap-
proximately 2700 cores, but the execution time for P-CSI is relatively flat beyond that core count regardless of preconditioner (Fig. 13). Using the EVP preconditioner, P-CSI can accelerate the barotropic calculation from 19.0 s to 4.4 s per simulation day on 16,875 cores. Dennis et al. (2012) indicated that 5 simulated years per wall-clock day is the minimum requirement to run long-term climate simulations. For the completed POP simulation, Fig. 14 indicates that the simulated tim-
ing of P-CSI achieves 10.5 simulated years per wall-clock day on 16,875 cores, whereas the timing of ChronGear with a diagonal preconditioner achieves only 6.2 simulated years per wall-clock day using the same number of cores. In Section 2, we demonstrated that the percentage of the POP execution time required by the barotropic solver increases with increasing number of cores using the original ChronGear solver. In particular, ChronGear with diagonal preconditioning accounts for ap-
proximately 50% of the total execution time on 16,875 cores (see Fig. 3). In contrast, Fig. 15 shows that by using the scalable P-CSI solver, the barotropic calculation time constitutes only approximately 16% of the total execution time on 16,875 cores. Finally, note that verification results for 1° POP by an ensemble-based statistical method in Hu et al. (2015) indicate that our new solver does not introduce statistically significant error into the simulation results.

## 6 Conclusions

We accelerated the high-resolution POP in the CESM framework by implementing a new P-CSI ocean barotropic solver. This new solver adopts a Chebyshev-type iterative method to avoid the global communication operations in conjunction with an effective EVP preconditioner to improve the parallel performance further. The superior performance of the P-CSI is carefully investigated



using the theoretical analysis of the algorithm and computational complexity. Compared with the existing ChronGear solver, it significantly reduces the global reductions and realizes a competitive convergence rate. The proposed alternative has become the default barotropic solver in the POP within CESM and may greatly benefit other climate models.

## 7 Code availability

The present P-CSI solver v1.0 is available on https://zenodo.org/record/56705 and https://github.com/ hxmhuang/PCSI. This solver is also included in the upcoming CESM public release v2.0. For the older CESM versions 1.2.x, the user should follow these steps indicated in the Readme.md file:

(1) Create a complete case or an ocean component case.

(2) Copy our files into the corresponding case path and build this case.

(3) Add two lines at the end of user_nl_pop2 file to use our new solver.

(4) Execute the preview_namelists file to activate the solver.

(5) Run the case.

The user are welcome to see the website mentioned above for more details and use the configuration files to repeat our experiments.





## Appendix A: Algorithms

### A1   PCG algorithm

The procedure of PCG is shown as follows (Smith et al., 2010):

Initial guess: $\mathbf{x}_0$

Compute residual $\mathbf{r}_0 = \mathbf{b} - \mathbf{A}\mathbf{x}_0$

Set $\mathbf{s}_0 = 0$, $\beta_0 = 1$

For $k = 1, 2, \cdots, k_{max}$ do

1. $\mathbf{r}'_{k-1} = \mathbf{M}^{-1}\mathbf{r}_{k-1}$

2. $\beta_k = \mathbf{r}^T_{k-1}\mathbf{r}'_{k-1}$

3. $\mathbf{s}_k = \mathbf{r}'_{k-1} + (\beta_k/\beta_{k-1})\mathbf{s}_{k-1}$

4. $\mathbf{s}'_k = \mathbf{A}\mathbf{s}_k$

5. $\alpha_k = \beta_k/(\mathbf{s}^T_k\mathbf{s}'_k)$

6. $\mathbf{x}_k = \mathbf{x}_{k-1} + \alpha_k\mathbf{s}_k$

7. $\mathbf{r}_k = \mathbf{r}_{k-1} - \alpha_k\mathbf{s}'_k$

8. convergence_check($\mathbf{r}_k$)

End Do

Operations such as $\beta_k/\beta_{k-1}$ in line (3) are scaler computations, whereas $\alpha_k\mathbf{s}_k$ in line (6) are vector scalings. $\mathbf{A}\mathbf{s}_k$ in line (4) is a matrix-vector multiplication. Inner products of vectors are $\mathbf{r}^T_{k-1}\mathbf{r}'_{k-1}$ in line (2) and $\mathbf{s}^T_k\mathbf{s}'_k$ in line (5).

### A2   ChronGear algorithm

The procedure of ChronGear is shown as follows:

Initial guess: $\mathbf{x}_0$

Compute residual $\mathbf{r}_0 = \mathbf{b} - \mathbf{A}\mathbf{x}_0$

Set $\mathbf{s}_0 = 0$, $\mathbf{p}_0 = 0$, $\rho_0 = 1$, $\sigma_0 = 0$

For $k = 1, 2, \cdots, k_{max}$ do

1. $\mathbf{r}'_k = \mathbf{M}^{-1}\mathbf{r}_{k-1}$

2. $\mathbf{z}_k = \mathbf{A}\mathbf{r}'_k$






   3. $\rho_k = \mathbf{r}_{k-1}^T \mathbf{r}_k'$

   4. $\sigma_k = \mathbf{z}_k^T \mathbf{r}_k' - \beta_k^2 \sigma_{k-1}$

   5. $\beta_k = \rho_k / \rho_{k-1}$

   6. $\alpha_k = \rho_k / \sigma_k$

   7. $\mathbf{s}_k = \mathbf{r}_k' + \beta_k \mathbf{s}_{k-1}$

8. $\mathbf{p}_k = \mathbf{z}_k + \beta_k \mathbf{p}_{k-1}$

   9. $\mathbf{x}_k = \mathbf{x}_{k-1} + \alpha_k \mathbf{s}_k$

   10. $\mathbf{r}_k = \mathbf{r}_{k-1} - \alpha_k \mathbf{p}_k$

   11. convergence_check($\mathbf{r}_k$)

End Do


### A3  P-CSI algorithm

The pseudocode of the P-CSI algorithm is shown as follows:

Initial guess: $\mathbf{x}_0$, estimated eigenvalue boundaries $[\nu, \mu]$

Set $\alpha = \frac{2}{\mu-\nu}$, $\beta = \frac{\mu+\nu}{\mu-\nu}$, $\gamma = \frac{\beta}{\alpha}$, $\omega_0 = \frac{2}{\gamma}$

Compute residual $\mathbf{r}_0 = \mathbf{b} - \mathbf{A}\mathbf{x}_0$, $\Delta\mathbf{x}_0 = \gamma^{-1}\mathbf{M}^{-1}\mathbf{r}_0$, $\mathbf{x}_1 = \mathbf{x}_0 + \Delta\mathbf{x}_0$, $\mathbf{r}_1 = \mathbf{b} - \mathbf{A}\mathbf{x}_1$

For $k = 1, 2, \cdots, k_{max}$ do

   1. $\omega_k = 1/(\gamma - \frac{1}{4\alpha^2}\omega_{k-1})$

   2. $\mathbf{r}_k' = \mathbf{M}^{-1}\mathbf{r}_k$

3. $\Delta\mathbf{x}_k = \omega_k \mathbf{r}_k' + (\gamma\omega_k - 1)\Delta\mathbf{x}_{k-1}$

   4. $\mathbf{x}_{k+1} = \mathbf{x}_k + \Delta\mathbf{x}_k$

   5. $\mathbf{r}_{k+1} = \mathbf{b} - \mathbf{A}\mathbf{x}_{k+1}$

   6. convergence_check($\mathbf{r}_k$)

End Do






### Appendix B: Eigenvalue Estimation

The procedure of the Lanczos method to estimate the extreme eigenvalues of the matrix $M^{-1}A$ is shown as follows:

Initial guess: $\mathbf{r}_0$

Set $\mathbf{s}_0 = \mathbf{M}^{-1}\mathbf{r}_0$; $\mathbf{q}_1 = \mathbf{r}_0/(\mathbf{r}_0^T\mathbf{s}_0)$; $\mathbf{q}_0 = \mathbf{0}$; $\beta_0 = 0$;    $\mu_0 = 0$; $T_0 = \emptyset$

For $j = 1, 2, \cdots, m$ do

1. $\mathbf{p}_j = \mathbf{M}^{-1}\mathbf{q}_j$

2. $\mathbf{r}_j = \mathbf{A}\mathbf{p}_j - \beta_{j-1}\mathbf{q}_{j-1}$

3. $\alpha_j = \mathbf{p}_j^T\mathbf{r}_j$

4. $\mathbf{r}_j = \mathbf{r}_j - \alpha_j\mathbf{q}_j$

5. $\mathbf{s}_j = \mathbf{M}^{-1}\mathbf{r}_j$

6. $\beta_j = \mathbf{r}_j^T\mathbf{s}_j$

7. **if**   $\beta_j == 0$   **then   return**

8. $\mu_j = max(\mu_{j-1}, \alpha_j + \beta_j + \beta_{j-1})$

9. $T_j = tri\_diag(T_{j-1}, \alpha_j, \beta_j)$

10. $\nu_j = eigs(T_j, 'smallest')$

11. **if** $|\frac{\mu_j}{\mu_{j-1}} - 1| < \epsilon$   **and**   $|1 - \frac{\nu_j}{\nu_{j-1}}| < \epsilon$   **then   return**

12. $\mathbf{q}_{j+1} = \mathbf{r}_j/\beta_j$

End Do

In step (9), $T$ is a tridiagonal matrix which contains $\alpha_j (j = 1, 2, ..., m)$ as the diagonal entries and $\beta_j (j = 1, 2, ..., m-1)$ as the off-diagonal entries.

$$
T_m = \begin{bmatrix}
\alpha_1 & \beta_1 & & & \\
\beta_1 & \alpha_2 & \beta_2 & & \\
 & \beta_2 & \ddots & \ddots & \\
 & & \ddots & \ddots & \beta_{m-1} \\
 & & & \beta_{m-1} & \alpha_m
\end{bmatrix}
$$

Let $\xi_{min}$ and $\xi_{max}$ be the smallest and largest eigenvalues of $T_m$, respectively. Paige (1980) demonstrated that $\nu \leq \xi_{min} \leq \nu + \delta_1(m)$ and $\mu - \delta_2(m) \leq \xi_{max} \leq \mu$. As $m$ increases, $\delta_1(m)$ and





$\delta_2(m)$ will gradually converge to zero. Thus, the eigenvalue estimation of $M^{-1}A$ is transformed to solve the eigenvalues of $T_m$. Step (8) in eigenvalue estimation employs the Gershgorin circle theorem to estimate the largest eigenvalue of $T_m$, that is, $\mu = \max_{1 \le i \le m} \sum_{j=1}^{m} |T_{ij}| = \max_{1 \le i \le m} (\alpha_i + \beta_i +$

$\beta_{i-1})$. The efficient QR algorithm (Ortega and Kaiser, 1963) with a complexity of $\Theta(m)$ is used to estimate the smallest eigenvalue $\nu$ in step (9).

*Acknowledgements.* This work is supported in part by a grant from the National Natural Science Foundation of China (41375102), and the National Grand Fundamental Research 973 Program of China (No. 2014CB347800). Computing resources were provided by the Climate Simulation Laboratory at NCAR's Computational and

Information Systems Laboratory (sponsored by the NSF and other agencies).



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





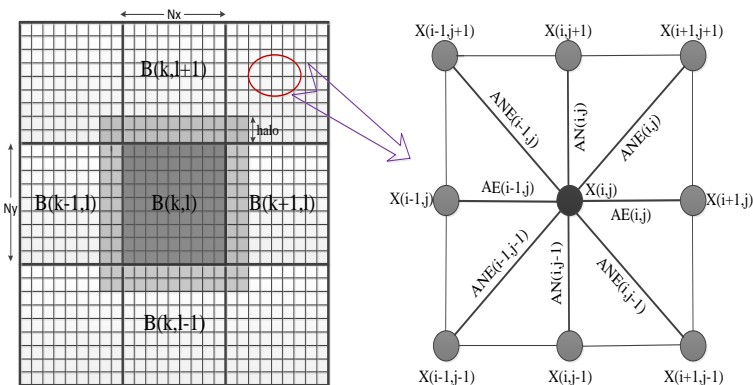

**Figure 1.** Grid domain decomposition of the ocean model component in CESM.





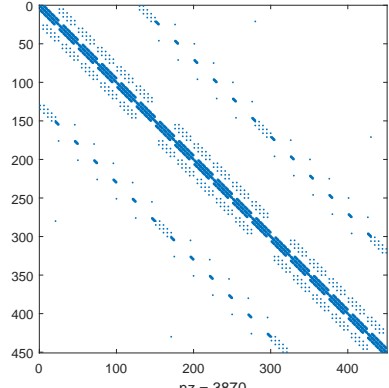

**Figure 2.** Sparse pattern of the coefficient matrix in the case with $30 \times 15$ grids using nine-point stencils.





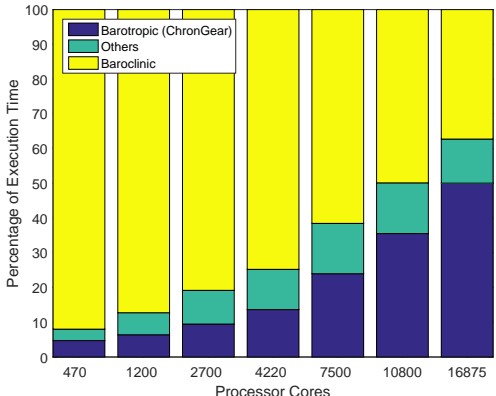

**Figure 3.** Percentage of execution time in 0.1° POP using the default diagonal-preconditioned ChronGear solver on Yellowstone.





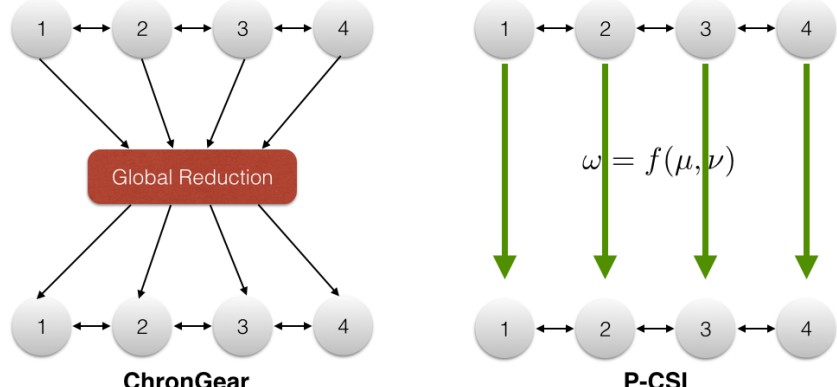

**Figure 4.** Workflow of ChronGear and P-CSI iterations when four processes are used.





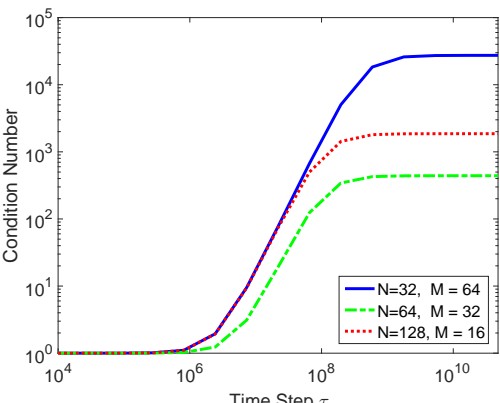

**Figure 5.** Relationship between time step size and the condition number of the coefficient matrix with fixed number of grid points. $N$ and $M$ are numbers of grid points along the perimeter and height of the cylinder.



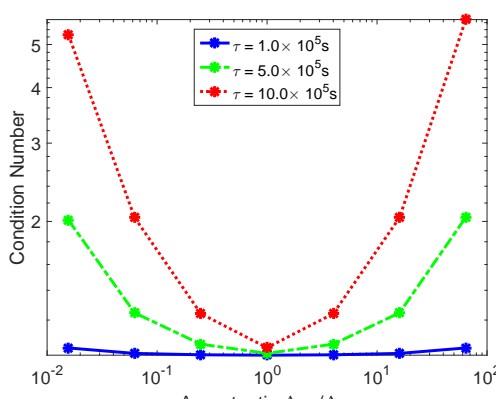

**Figure 6.** With fixed grid size ($\mathcal{N} = 2048$) and varies time step size $\tau$, relationship between aspect ratio and the condition number of the coefficient matrix.





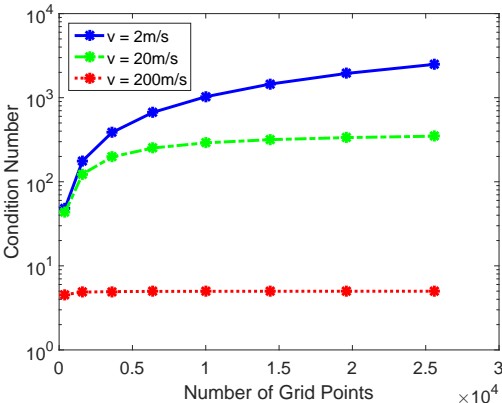

**Figure 7.** When the aspect ratio is constant, relationship between the number of grid points and the condition number of the coefficient matrix. The affect of time step sizes are tested by configuring with various supported barotropic velocities $v$.





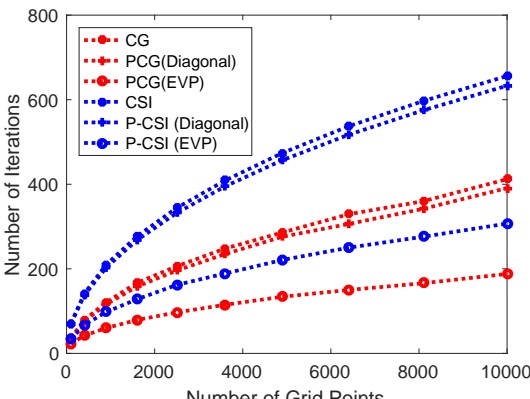

**Figure 8.** Relationship between grid sizes and number of iterations of different solvers in test cases with the idealized configuration.





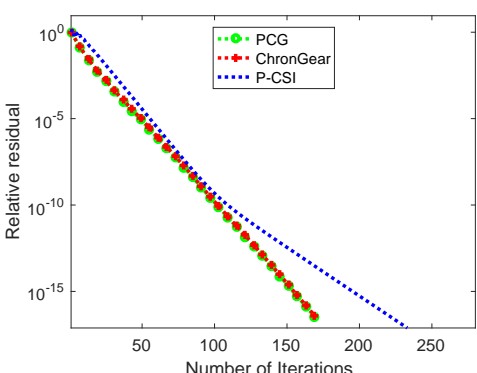

**Figure 9.** The convergence rate of different barotropic solvers in the 0.1° POP.





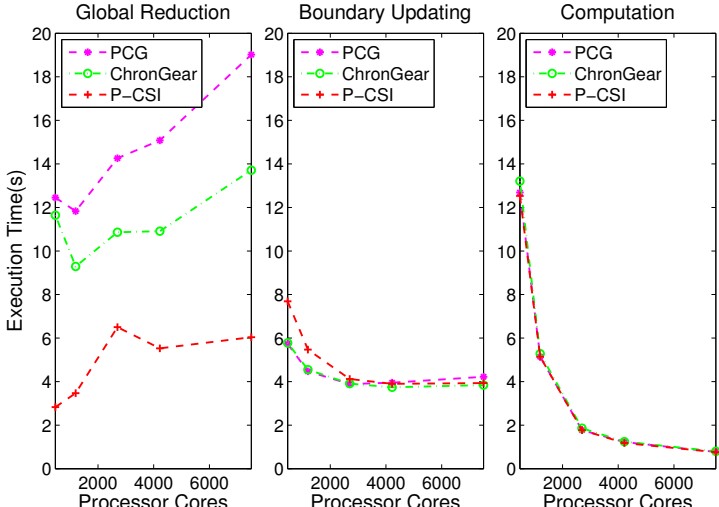

**Figure 10.** The execution time for different phases in the barotropic solvers in the 0.1° POP on Yellowstone.





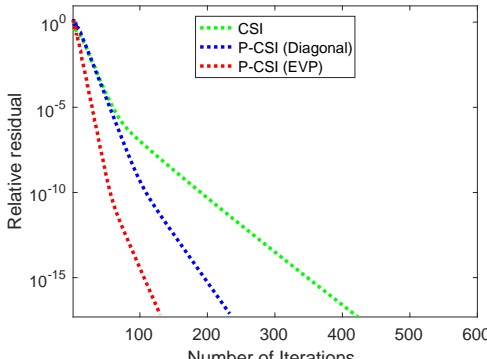

**Figure 11.** The convergence rate of P-CSI solver with different preconditioners in the 0.1° POP on Yellowstone.

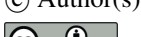



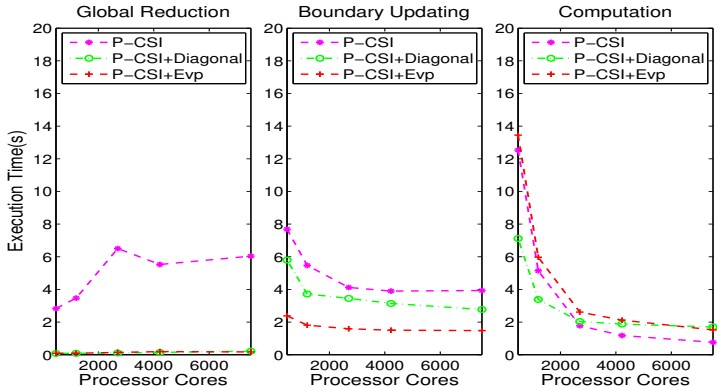

**Figure 12.** The execution time for different phases with different preconditioners in the P-CSI solvers.





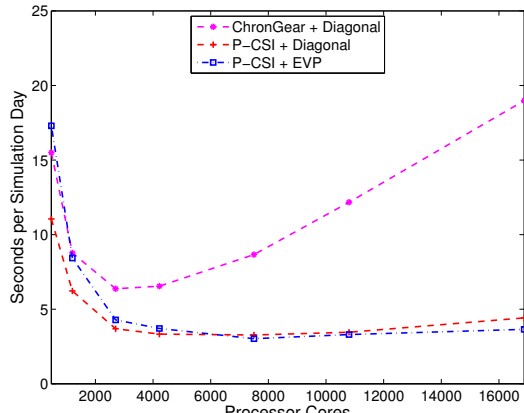

**Figure 13.** The execution time for P-CSI solver with different preconditioners in the 0.1° ocean model component for one simulation day on Yellowstone. Note that this figure is a subset of Fig. 8 in Hu et al. (2015)





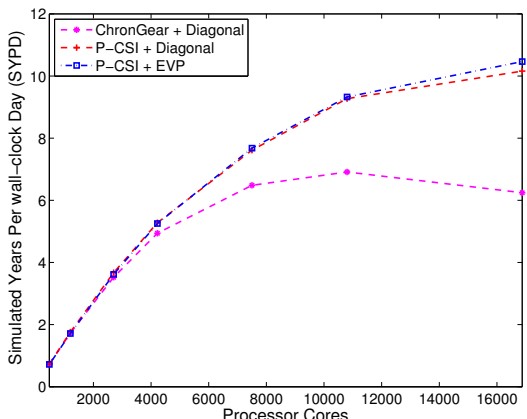

**Figure 14.** The simulated speed of the 0.1° ocean model component on Yellowstone. Note that this figure is a subset of Fig. 8 in Hu et al. (2015)





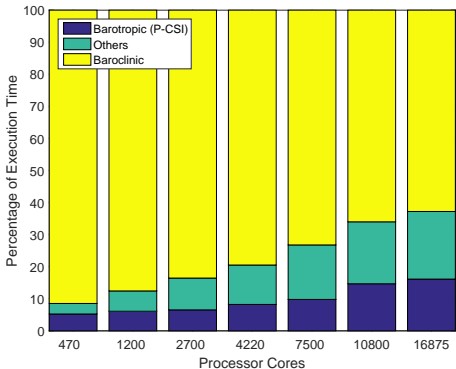

**Figure 15.** Percentage of execution time in the 0.1° POP using P-CSI. Note that this figure uses the same data as that in Fig. 9 in Hu et al. (2015)




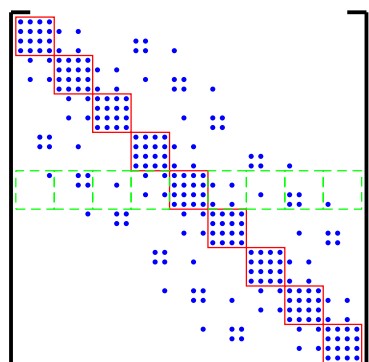

**Figure 16.** sparsity pattern of the coefficient matrix developed from nine-point stencils. the whole domain is divided into $3 \times 3$ non-overlapping blocks. elements in red rectangles are coefficients between points in blocks. elements in blue rectangles are coefficients between points from the $i$-th block and its neighbor blocks.





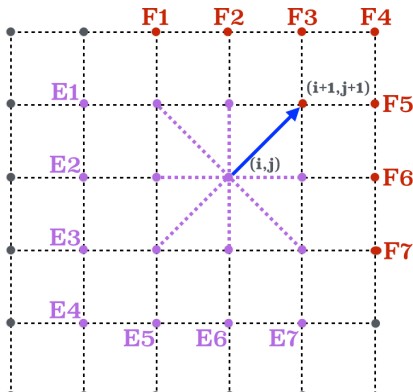

**Figure 17.** evp marching method for nine-point stencil. the solution on point $(i+1, j+1)$ can be calculated using the equation on point $(i,j)$, providing solutions on other neighbor points of point $(i,j)$.