# Peer review of "P-CSI v1.0, an accelerated barotropic solver for the high-resolution ocean model component in the Community Earth System Model v2.0"

_Geoscientific Model Development, 2016_

## Referee Comment (RC1) · E. Müller (Referee) · 25 Aug 2016

**1   General comments**

In their paper: "P-CSI v1.0, an accelerated barotropic solver for the high resolution ocean model component in the Community Earth System Model v2.0" the authors discuss the implementation of a new preconditioned iterative solver algorithm for the solution of the elliptic PDE which arises in implicit time stepping of the barotropic mode in ocean models. By using an iterative method which avoids global communications,

the scalability of the solver can be improved significantly on large core counts. The authors demonstrate that this leads to substantial performance improvements when running the model at high spatial resolution on more than 16,000 cores of the Yellowstone supercomputer.

Accurate and scalable models are absolutely essential for reliable predictions of the Earth's climate which have wide impact in the geoscience community. In the introduction the authors argue convincingly why the development of high resolution ocean models and of massively parallel elliptic solvers is necessary and their novel algorithm approach addresses an important bottleneck for scalability on large core counts (global parallel reductions). Since the barotropic solver accounts for a large fraction in the runtim, this work has a large impact for the numerical model they study and can also help to improve related models in atmospheric- and ocean- modelling. The work is put into context by referring to relevant related publications and the paper is very well written throughout, with the results supporting the theoretical analysis (in particular the theoretical performance analysis). The scientific results, in particular the use of a communication-avoiding iterative method as an alternative to "standard" Krylov-subspace methdos such as CG, are very interesting and the benefits of the method are demonstrated convincingly by detailed numerical experiments.

As stated at the end of the introduction, this paper is based on a related conference proceedings publication [1], where the key ingredients of the algorithm and parallel scaling tests are described in detail. Compared to [1], the present GMD paper contains the following new material:

1. The properties of the discretised system and in particular the spectral radius of the matrix is derived for a simplified test setup (constant ocean depth). By a theoretical analysis of the convergence rate of the new P-CSI algorithm the authors argue that it converges as fast as the CG and ChronGear solvers which require additional global reductions

2. Numerical estimates of the spectral radius for different aspect ratios and time step sizes are presented

3. The detailed convergence history of different solvers/preconditioners is studied

However, the key concepts of the solver/preconditioner setup and similar results are already given in [1] (for example for the convergence numbers, compare Fig. 6 in [1] and Figs. 9 11 in the present paper, Fig. 13-15 are a subset of results from [1] and as far as I can tell Fig. 10 and 12 are obtained with a similar setup as in [1]). I'm therefore slightly concerned as to whether this paper contains sufficient new material to for a new publication, in particular since I'm not sure how relevant the variations in time step size really are in practice - the time step size is largely fixed by the CFL limit in other model components, such as the advective time scale (see further discussion below). To publish the paper it has to be made clear that large parts of it consist of new results.

The solver code is made available online, but I was not able to compile and run it since it requires installation of the full model.

**2 Specific comments**

- Early on in the introduction and in section 2.2 the authors mention that global reductions limit the performance of ChronGear and CG solvers, and this is one of the main motivations for using the P-CSI method. While this is clearly shown in Fig. 10, it might be good to already refer to numerical evidence here or quote numbers from [1]: which percentage of the runtime is spent in global reduction operations?

- In section 2 the authors derive the barotropic mode in the fundamental equations and then discretise it implicitly in time. At this point it might be good to briefly

mention how this is related to other model components: how is the implicitly calculated height perturbation coupled back to the full equations? Are other parts of the equations (such as advection) solved explicitly? What are the typical time velocities/time scales (I assume that the implicit treatment of the barotropic mode is necessary since the gravity-wave velocity $\sqrt{g * H}$ is much larger than other velocities in the system, such as advective velocities - is this correct?).

- When showing strong scaling results such as in Figs. 3, 13-15 the number of unknowns per processor is relevant to assess the relative importance of halo exchange, could this information be added to the figures?

- Discussion in section 4.1: the barotropic CFL number due to gravity waves of speed $c_g = \sqrt{gH}$, $n_g = c_g \cdot dt/dx$, is a very important quantity and for a given resolution directly related to the time step size. However, typically the time step size $dt$ is limited by other processes in the model. For example, if there is another process with typical speed c', which is treated explicitly, then the related CFL number $n' = c' \cdot dt/dx$ is limited by $n' < O(1)$. For example in atmospherical models $c_{advection}\ 10 \cdot c_{acoustic}$, and hence the $n_g$ should not be larger than $\approx 10$. Could the authors include a discussion of this and also discuss physical limits on $n_g$ by referring to other components (i.e. non-barotropic) of the model? I think this is very important since the CFL number has a significant impact on the solver performance. By using $c_g = \sqrt{9.81m/s^2 \times 4km} = 200m/s$ the large time step sizes in Fig. 5 seems to be completely unphysical if I assume that there is another explicitly treated process in the model which is $\approx 10\times$ slower than the gravity waves, but my intuition from atmospheric models might be misleading here and if the explicitly treated non-barotropic dynamics happens at much larger time scales then those large time steps make sense. This seems to be implied by the setup used for the 0.1 degree runs: assuming a depth of 4km, a time step size of 172.8s would lead to a CFL number of $\approx 10^4$. Since the condition number depends largely on the CFL number it would be good to see what the physically

relevant values are.

- The condition $dt = dx/v$, which is imposed at the bottom of page 9 should be clarified. The authors refer to $v$ as the "barotropic velocity" and then vary this between 2m/s and 200m/s. Should this v be some other velocity in the system which limits the time step size? The velocity relevant for the barotropic equation is the gravity wave speed $c_g = \sqrt{gH}$ which is 200m/s for a depth of 4km. Same question in section 5.1, where the authors fix $v = 2m/s$.

- It would help if the CFL number and (an estimate of the) condition number of the matrix are given for the realistic 0.1 degree run. Since the largest and smallest eigenvalue are estimated this information should be available.

- If the CFL numbers are very large (see previous points), then I really think that advanced preconditioners have the potential for improving the performance. Multigrid preconditioners could reduce the iteration count from $O(100)$ to $O(10)$, so might pay off even if one preconditioner application is more expensive.

- In Fig. 5 it would be good to indicate the range of typical physical time step sizes for each resolution instead of just plotting a wide range of time scales.

- Page 13, line 394: while the matrix becomes more ill conditioned as the problem size increases, the condition $dt = dx/v$ will limit this growth, in fact the upper bound on the condition number is at the order of $\approx gH/v^2$.

- The theoretical analysis is carried out for a constant ocean depth $H$. How reasonable is this assumption and which impact do variations in $H$ have?

**3 typos/minor comments**

– at several places in the paper "scaler" should be replaced by "scalar"

- to me "boundary communication" is a slightly unusual expression, I'd call this "halo exchange" since "boundary" could refer to a physical boundary in the global domain (such as the ocean-land interface).

- at the bottom of page 4: should this read "[...] the barotropic continuity Eq. (4) *has been* linearised [...]" ("is linearised" implies that another term has to be removed from (4) to obtain a linear equation, but (4) is already linear).

- bottom of page 8: "spectrum radius" -> "spectral radius"

- definition of $P_k(\xi)$ between Eqs. (19) and (20) on page 10: What are $\alpha$ and $\beta$ here?

- in appendix A and B it might help if the global reduction operations in steps 2. and 5. of PCG and steps 3. and 4. of ChronGear are highlighted. Also, a sentence to the appendix which clarifies that the global reduction of $\rho_k$ and $\sigma_k$ in the ChronGear algorithm can be combined (thus halving the latency) might help.

- Fig. 4 does not add relevant information and should be removed

- Fig. 2: replace "sparse pattern" -> "sparsity pattern"

**references**

[1] Hu et al., 27th International Conference for High Performance Computing, Networking, Storage 395 and Analysis (SC2015), 2015

---

## Referee Comment (RC2) · Anonymous Referee #2 · 30 Aug 2016

**1   General comments**

This paper presents a new solver and preconditioner for the barotropic mode of the POP ocean model, focusing on parallel performance.  The new CSI solver is slower than the previous CG solver in serial, but it removes a parallel reduce operation which makes it significantly faster at high number of processes. The new EVP preconditioner reduces the number of iterations required by the solver, thus reducing both computation and communication time.  The authors demonstrate that the new P-CSI solver significantly improves the efficiency of the POP model in massively parallel runs.

The paper extends an earlier proceedings paper (Hu et al. 2015) by presenting: a more detailed description of the barotropic solver used in POP, analysis of the eigenvalues and condition number of the associated linear operator, analysis of the convergence rates of different solvers, and new estimates of the computational complexity of the solvers. It is however debatable whether the additional material merits another publication as most of the material originates from Hu et al. (2015). In addition the analysis of the condition numbers could be improved.

**2 Specific comments**

Section 4.1 Spectrum and condition number

As we are dealing with a 2D shallow water solver, this analysis would be clearer if it incorporated the non-dimensional barotropic CFL number. It would be useful to know what the CFL number of the 2D mode is in typical CESM runs, and use that as a basis for the analysis and idealized experiments.

Noting that $CFL = c\Delta t/\Delta x$, with $c = \sqrt{gH}$ being the speed of the gravity waves, $\phi$ in eq. (14) (and consequently the condition number) can be expressed as a function of CFL. Specifically $\phi = 1/(CFL_x CFL_y)$ and (using $\lambda_{min}, \lambda_{max}$ from line 273) the condition number is approximately $\kappa \approx 4CFL^2 + 1$ for large time steps and aspect ratio=1, which shows the dependency clearly.

Similarly in Figs 5, 6 and 7 it would be useful to use the CFL number instead of the time step or 2D velocity. The value of the time step alone, for example, is not informative as it depends on how the idealized run was set up.

Section 3.2 A block EVP preconditioner

In the last paragraph the authors mention that the drawback of the EVP preconditioner is that it cannot be used to solve large problems due to propagation of errors. Does

this imply that EVP cannot be used at low processor counts? If so have the authors experimented with or documented the failure of EVP? Is it possible to derive a threshold problem size under which the EVP preconditioner is reliable?

Section 4.3 Computational complexity

These estimates of computational complexity are similar to those presented in Hu et al. (2015). Some of them are different however (e.g. eqns. 26, 27, 28) especially in terms of the computation time $T_c$. Why the difference?

line 400: Fig. 8 shows that the P-CSI solver converges slower compared to PCG. Why is it so? The analysis in Section 4.2 concluded that the convergence rate should be similar.

line 415: It is not clear how the grid is divided in blocks. It might be worth explaining this is Section 2.1.

**3   Technical corrections**

line 109: $\rho_0$ is the constant reference density, not the actual water density

line 122: Meaning of the last sentence is unclear, please elaborate/reformulate.

line 170: typo: scalar

line 285: Here the authors assume that the time step satisfies the CFL condition, i.e. CFL<=1, where the velocity $v$ is chosen rather arbitrarily. It would be better to use CFL numbers typical to CESM applications.

fig 7: what is the grid aspect ratio used in this test?

line 335: $T_c$ is not defined in this paper

line 513: typo: scalar

figs. 16 and 17: these figures are not mentioned in the manuscript

---

## Author Comment (AC1) · 7 Sep 2016

Dear Dr. Müller:
We would like to express our sincere appreciation to your valuable feedback. Your comments are highly insightful and enable us to significantly improve our manuscript. The following pages are our point-by-point responses to each of your comments.

[Figure]

**1 Response to general comments**

In their paper: "P-CSI v1.0, an accelerated barotropic solver for the high resolution ocean model component in the Community Earth System Model v2.0" the authors discuss the implementation of a new preconditioned iterative solver algorithm for the solution of the elliptic PDE which arises in implicit time stepping of the barotropic mode in ocean models. By using an iterative method which avoids global communications, the scalability of the solver can be improved significantly on large core counts. The authors demonstrate that this leads to substantial performance improvements when running the model at high spatial resolution on more than 16,000 cores of the Yellowstone supercomputer.

Accurate and scalable models are absolutely essential for reliable predictions of the Earth's climate which have wide impact in the geoscience community. In the introduction the authors argue convincingly why the development of high resolution ocean models and of massively parallel elliptic solvers is necessary and their novel algorithm approach addresses an important bottleneck for scalability on large core counts (global parallel reductions). Since the barotropic solver accounts for a large fraction in the runtime, this work has a large impact for the numerical model they study and can also help to improve related models in atmospheric- and ocean- modelling. The work is put into context by referring to relevant related publications and the paper is very well written throughout, with the results supporting the theoretical analysis (in particular the theoretical performance analysis). The scientific results, in particular the use of a communication-avoiding iterative method as an alternative to "standard" Krylov subspace methods such as CG, are very interesting and the benefits of the method are demonstrated convincingly by detailed numerical experiments. As stated at the end of the introduction, this paper is based on a related conference proceedings publication [1], where the key ingredients of the algorithm and parallel scaling tests are described in detail. Compared to [1], the present GMD paper contains the following new material:

1.The properties of the discretized system and in particular the spectral radius of the matrix is derived for a simplified test setup (constant ocean depth). By a theoretical analysis of the convergence rate of the new P-CSI algorithm the authors argue that it converges as fast as the CG and ChronGear solvers which require additional global reductions.

2.Numerical estimates of the spectral radius for different aspect ratios and time step sizes are presented.

3.The detailed convergence history of different solvers/preconditioners is studied.

However, the key concepts of the solver/preconditioner setup and similar results are already given in [1] (for example for the convergence numbers, compare Fig. 6 in [1] and Figs. 9 11 in the present paper, Fig. 13-15 are a subset of results from [1] and as far as I can tell Fig. 10 and 12 are obtained with a similar setup as in [1]). I am therefore slightly concerned as to whether this paper contains sufficient new material to for a new publication, in particular since I'm not sure how relevant the variations in time step size really are in practice - the time step size is largely fixed by the CFL limit in other model components, such as the advective time scale (see further discussion below). To publish the paper it has to be made clear that large parts of it consist of new results. The solver code is made available online, but I was not able to compile and run it since it requires installation of the full model.

[Response]:
Thank you for your highly valued comments that our study has a large impact for the numerical model development and can help to improve related models in atmospheric and oceanic modelling.

This paper is an extended work originally presented in the 27th International Conference for High Performance Computing, Networking, Storage and Analysis (SC 2015) as we indicated in Section I. Most of the audiences in the SC conference are supercomputing specialists. Therefore, we simplified the background of the ocean model and focused on the design of algorithm, scalability tests and efficiency in the SC paper. We introduced our solver with some pseudo-code and used a lot of computer terminologies since the main readers are computer scientists. In order to expand the influence of our work to more general readers, we decide to extend our paper to GMD which is an outstanding academic exchange platform for climate modelers. Therefore, we made a lot of changes to the content and structure of the GMD manuscript comparing with the SC paper. Although you have mentioned part of them in the general comments, we summarize our major changes as follows:

(1) We enriched the review of barotropic mode and introduced two solvers including PCG and ChronGear adopted in the original POP. We believe that this important part will help other climate modelers to comprehensively understand a general large-scale computing problem in POP, MOM, MITgcm, FVCOM, OPA models etc. They can associate their own climate models with our new solver through the detailed barotropic equations. In SC paper, we only give a simple version of barotropic mode and the ChronGear solver.

(2) We combined sections 3 and 4 in SC paper into the section "Design of the P-CSI solver" in GMD manuscript. In order to make the climate modelers instead of computer scientists to better understand the new solver, we rewrote most of sentences and moved the pseudo-code and the procedure of preconditioned into the appendix for interested readers. We also avoided the use of obscure computer jargon to make it more readable.

(3) After our presentation at SC conference last year, many helpful advices were gathered. Some specialists pointed out that we should provide more information about the universal applicability of our new solver in different situations and different applications. Therefore, in the GMD manuscript, a new section 4 is dedicated to the characteristics of the new approach. We theoretically analyzed the characteristics of P-CSI through the associated eigenvalues and their connection with the convergence rate. Based on different solvers/preconditioners, we derived the properties of the discretized system and, in particular, the spectral radius of the matrix by theoretical analysis of the Spectrum, condition number and the convergence rate. We concluded that P-CSI can converge as fast as the CG and ChronGear solvers which require additional global reductions. In SC paper, we only presented the computational complexity.

We agree that there are some similar materials in the GMD manuscript and SC paper in section 5 because this section presents the actual computational evaluation about the actual performance. We will make some adjustments as follows:

(1) Fig. 9 is pretty new (not in the SC paper) and verifies the theoretical analysis of the convergence rate of different barotropic solvers, we will merge it with Fig. 11 which is also new and confirms the improvement of convergence rate. We will add some slopes on Fig. 11 to reflect the condition numbers improved in the different pre-conditioners.

(2) The material we discuss about Figs. 10, and 12 is very similar to that in SC paper. I believe it is still very important to emphasize that the global reduction is the major bottleneck in the large-scale computing ($>O(10^3)$ processor counts) in the GMD paper. The P-CSI with EVP preconditioner enhances the performance by significantly reducing the iteration number so that the time for computation part is further reduced. In the revised version, we will further combine Fig. 10 and

12 and reemphasize this point by shortening the discussion and an overall comparison.

(3) We will eliminate Fig. 13 because it illustrates the problem of ChronGear + Diagonal and doesn't add much. Figs. 14 and 15 will be reduced into one figure. In the SC paper, we presented the analysis for barotropic solver only. Here, we will extend to the overall performance of model by including the timing for other components. This information is very important because the performance of 1/10 degree POP is not clearly documented yet. So, we will put the total time (all components). This will make a big difference and show the major advantage of our new approach.

According to your concern about the unphysical time step size in Fig.5, we will use a typical value for the first baroclinic wave speed of 2m/s and a typical gravity wave velocity of 200m/s to reanalyze the solver property. In the 0.1 degree ocean simulation, the CFL number is about $c\Delta t/\Delta x \approx 3.46$, thus the corresponding time step size are 17280s and 172.8s. We also believe it will make more senses to use CFL number as the x-axis in some plots (see the following responses) instead of time-step (which is also connected by dx). Therefore, all relevant figures will be changed accordingly.
In conclusion, after making the above modifications, we believe that this GMD manuscript will be an expanded, more fully developed, and more refined version of the conference paper. These new results are confirmed by a valuable theoretical analysis. All figures have completely different stories and key messages comparing the SC paper. We will clearly introduce these differences in the revised manuscript if we have the opportunity.
**2 Response to specific comments**

(1) Early on in the introduction and in section 2.2 the authors mention that global reductions limit the performance of ChronGear and CG solvers, and this is one of the main motivations for using the P-CSI method. While this is clearly shown in Fig. 10, it might be good to already refer to numerical evidence here or quote numbers from [1]: which percentage of the runtime is spent in global reduction operations?

[Response]:

In the introduction, we will add the following sentence,

"For example, when around four thousand cores are used, the global reduction in PCG and ChronGear takes approximately 74% and 68% of the whole barotropic time, respectively [1]. This situation will get worse with more cores."

Also, Fig. 10 will be merged with Fig. 12 so that we can reemphasize the advantage of our proposed method. This discussion will be shortened.

(2) In section 2 the authors derive the barotropic mode in the fundamental equations and then discretise it implicitly in time. At this point it might be good to briefly mention how this is related to other model components: how is the implicitly calculated height perturbation coupled back to the full equations? Are other parts of the equations (such as advection) solved explicitly? What are the typical time velocities/time scales(I assume that the implicit treatment of the barotropic mode is necessary since the gravity-wave velocity $\sqrt{g * H}$ is much larger than other velocities in the system, such as advective velocities - is this correct?).

[Response]:

Thanks for your good suggestion. In the beginning of section 2.1, we will briefly introduce the following time scheme.

"POP uses the splitting technique to solve the barotropic and baroclinic system [1]. All terms in the Eq. (1) use the explicit scheme except the implicit treatment of barotropic mode and semi-implicit treatment of Coriolis and vertical mixing terms. The implicit

treatment of barotropic mode is necessary to simulate the fast gravity waves with the speed of $\sqrt{g*H}$ ≈200m/s so that we can use the same time step as the baroclinic mode which has a velocity scale less than 2 m/s [1]. The time step of the 0.1 degree POP model is 172.8s."
Yes, you are correct that the implicit approach is necessary to match the same time step between the barotropic and baroclinic modes. Hope the above brief discussion can give the readers an overall description.

(3) When showing strong scaling results such as in Figs. 3, 13-15 the number of unknowns per processor is relevant to assess the relative importance of halo exchange, could this information be added to the figures?
[Response]:
This advice is valuable and we will add this information to Fig. 3 by including the new axis labeled as "number of grids per core" . We will eliminate Fig.13 because it just says the problem of ChronGear + Diagonal. The Figs. 14 and 15 will be reduced into one figure. We will refer to Fig. 3 for this information.

(4) Discussion in section 4.1: the barotropic CFL number due to gravity waves of speed $c_g = \sqrt{gH}, n_g = c_g dt/dx$, is a very important quantity and for a given resolution directly related to the time step size. However, typically the time step size is limited by other processes in the model. For example, if there is another process with typical speed $c^{'}$, which is treated explicitly, then the related CFL number $n^{'} = c^{'} dt/dx$ is limited by $n^{'} < O(1)$. For example in atmospherical models $c_{advection} 10 \cdot c_{acoustic}$, and hence the $n_g$ should not be larger than ≈ 10. Could the authors include a discussion of this and also discuss physical limits on $n_g$ by referring to other components (i.e. non-barotropic) of the model? I think this is very important since the CFL number has a significant impact on the solver performance. By using $c_g = \sqrt{9.81m/s^2 \times 4km} = 200m/s$ the large time step sizes in Fig. 5 seems to be completely unphysical if I assume that there is another explicitly treated process in the

model which is $\approx 10\times$ slower than the gravity waves, but my intuition from atmospheric models might be misleading here and if the explicitly treated non-barotropic dynamics happens at much larger time scales then those large time steps make sense. This seems to be implied by the setup used for the 0.1 degree runs: assuming a depth of 4km, a time step size of 172.8s would lead to a CFL number of $\approx 10^4$. Since the condition number depends largely on the CFL number it would be good to see what the physically relevant values are.

[Response]:

Thanks for your corrections. Considering the limitation of CFL condition, the time step size in Fig. 5 is beyond the physical range indeed. The original purpose of this selection is to make it more intuitive to readers that the time step sizes have a large influence on the condition number of the coefficient matrix without taking the physical consistency into account. We will redraw Fig.5 with the CFL number as the x-axis and make the values more credible in physics. In 0.1 degree ocean simulation, the time step size is $\Delta t$=172.8s (500 steps per simulation day), thus for the barotropic mode, the CFL number is about $c \cdot \Delta t / \Delta x \approx 3.46$ where $c = 200 m/s$ is the gravity wave velocity in the barotropic mode and $\Delta x = 10000 m$ is the horizontal grid length. If $c^{'} = 2m/s$ is a typical value for the first baroclinic wave speed, the CFL number is less than $c^{'} \cdot \Delta t / \Delta x \approx 0.035$. CFL numbers varying from 0.01 to 5 will be used in Fig. 5 to cover more physically relevant cases in POP.

(5) The condition $dt = dx/v$, which is imposed at the bottom of page 9 should be clarified. The authors refer to $v$ as the "barotropic velocity" and then vary this between 2m/s and 200m/s. Should this v be some other velocity in the system which limits the time step size? The velocity relevant for the barotropic equation is the gravity wave speed $c_g = \sqrt{gH}$ which is 200m/s for a depth of 4km. Same question in section 5.1, where the authors fix $v = 2m/s$.

[Response]:

Thanks for your corrections. We will use the typical value for the first baroclinic wave

speed 2m/s and the typical gravity wave velocity $c_g = \sqrt{gH} = 200m/s$ for a depth of 4km as the lower bound and the upper bound of the velocity range, respectively. Besides, we will use the non-dimensional barotropic CFL number instead of velocity as the legend of Fig. 7 to show the dependency.

(6) It would help if the CFL number and (an estimate of the) condition number of the matrix are given for the realistic 0.1 degree run. Since the largest and smallest eigenvalue are estimated, this information should be available.
[Response]:
We will add this information in Section 4.1.
"In 0.1 degree realistic run, the CFL number is about $c \cdot \Delta t/\Delta x \approx 3.46(c = 200m/s, \Delta t = 172.8s, \Delta x = 10000m$ is the typical gravity wave speed, time step and spatial resolution, respectively) and the condition number is about 250."

(7) If the CFL numbers are very large (see previous points), then I really think that advanced preconditioners have the potential for improving the performance. Multigrid preconditioners could reduce the iteration count from O(100) to O(10), so might pay off even if one preconditioner application is more expensive.
[Response]:
As you point out, any advanced preconditioner which can quickly reduce the iteration count will be very useful to improve the performance. In fact, the EVP solver is a direct fast solver so that it has this capability and is the main reason we choose here. Furthermore, as we indicated in lines 66-77 of section 1, the multigrid method is a well-known scalable and efficient approach to solve the elliptic systems too. However, some related works confirmed that the geometric multigrid in global ocean models does not always scale ideally because of the presence of complex topography (land particularly), non-uniform or anisotropic grids ([2], [3], [4], [5], [6]). These constraints lead to an elliptic system with variable coefficients defined on an irregular domain in POP and complicate the modeling system. The algebraic multigrid (AMG) is an

alternative to the geometric multigrid to handle the complex topography. However, the AMG setup in the parallel environment is more expensive than the iterative solver in climate modelling, which makes it unfavorable as a preconditioner [2]. On the contrary, the EVP preconditioner is simple enough and can effectively reduce the condition number of coefficient matrix by about 5 times in both 1 and 0.1 degree cases, which leads to a reduction of 2/3 iterations. Therefore, we use the direct EVP solver. Even so, more study about the preconditioner in practical climate models will be very useful and we will take it as our future work.

(8) In Fig. 5 it would be good to indicate the range of typical physical time step sizes for each resolution instead of just plotting a wide range of time scales.
[Response]:
We will replace the time step sizes in the x-axis of Fig. 5 with the non-dimensional CFL number and make the values more credible in physics.

(9) Page 13, line 394: while the matrix becomes more ill conditioned as the problem size increases, the condition $dt = dx/v$ will limit this growth, in fact the upper bound on the condition number is at the order of $\approx gH/v^2$.
[Response]:
Thanks for the corrections. We will add the following sentences in line 394 to make our revised version more rigorous.
"As shown in Fig. 8, when the problem size increases, the coefficient matrix becomes more poorly conditioned until it reaches the upper bound at the order of $gH/v^2$. "

(10) The theoretical analysis is carried out for a constant ocean depth H. How reasonable is this assumption and which impact do variations in H have?
[Response]:
The purpose of this assumption is to succinctly demonstrate the properties of the sparse matrix used in the POP. This assumption is not convincing in physics indeed,

while it provides a bound for cases with various H. We will expand our discussion to support variations in H, which should lead to similar results.

**3 Response to typos/minor comments**

(1) at several places in the paper "scaler" should be replaced by "scalar"
[Response]: We will replace "scaler" with "scalar" in our revised version.

(2) to me "boundary communication" is a slightly unusual expression, I'd call this "halo exchange" since "boundary" could refer to a physical boundary in the global domain (such as the ocean-land interface).
[Response]: We will replace "boundary communication" with "halo exchange" in our revised version.

(3) at the bottom of page 4: should this read " [...] the barotropic continuity Eq. (4) *has been* linearised [...]" ("is linearised" implies that another term has to be removed from (4) to obtain a linear equation, but (4) is already linear).
[Response]: We will change "is linearized" into "has been linearized" in our revised version.

(4) bottom of page 8: "spectrum radius" -> "spectral radius"
[Response]: We will revise "spectrum radius" with "spectral radius" in our revised version.

(5) definition of $P_k(\xi)$ between Eqs. (19) and (20) on page 10: What are $\alpha$ and $\beta$ here?
[Response]: We will introduce the meaning of $\alpha$ and $\beta$ in our revised version. The $\alpha$ is

the aspect ratio and $\beta$ is the reciprocal of $\alpha$ which are defined in Eq. (13).

(6) in appendix A and B it might help if the global reduction operations in steps 2. and 5. of PCG and steps 3. and 4. of ChronGear are highlighted. Also, a sentence to the appendix which clarifies that the global reduction of $\rho_k$ and $\sigma_k$ in the ChronGear algorithm can be combined (thus halving the latency) might help.

[Response]: We will highlight these steps and add some sentences to clarify the global reduction operations. In line 515, we will add "these inner products use two global reduction operations."; In line 535, we will add "The inner products in $\rho_k$ and $\sigma_k$ use two global reduction operations. However, these two global reductions can be combined into one operation thus halving the latency."

(7) Fig. 4 does not add relevant information and should be removed.
[Response]: We will remove Fig.4 and related paragraphs in our revised version.

(8) Fig. 2: replace "sparse pattern" -> "sparsity pattern"
[Response]: We will replace "sparse pattern" with "sparsity pattern" in our revised version.

**4 References**

[1] Hu et al., 27th International Conference for High Performance Computing, Networking, Storage 395 and Analysis (SC2015), 2015
[2] Müller, E. H. and Scheichl, R.: Massively parallel solvers for elliptic partial differential equations in numerical weather and climate prediction, Quarterly Journal of the Royal Meteorological Society, 140, 2608-2624, 2014.
[3] Matsumura, Y. and Hasumi, H.: A non-hydrostatic ocean model with a scalable

multigrid Poisson solver, Ocean Modelling, 24, 15-28, 2008.

[4] Kanarska, Y., Shchepetkin, A., and McWilliams, J.: Algorithm for non-hydrostatic dynamics in the regional oceanic modeling system, Ocean Modelling, 18, 143-174, 2007.

[5] Fulton, S. R., Ciesielski, P. E., and Schubert,W. H.: Multigrid methods for elliptic problems: A review, Monthly Weather Review, 114, 943-959, 1986.

[6] Stüben, K.: A review of algebraic multigrid, Journal of Computational and Applied Mathematics, 128, 281-309, 2001.

[7] Tseng, Y.-h. and Ferziger, J. H.: A ghost-cell immersed boundary method for flow in complex geometry, Journal of computational physics, 192, 593-623, 2003.

[8] Smith, R., et al. "The Parallel Ocean Program (POP) Reference Manual Ocean Component of the Community Climate System Model (CCSM) and Community Earth System Model (CESM)."ÂăRep. LAUR-01853Âă141 (2010).
* * *

---

## Author Comment (AC2) · 7 Sep 2016

Dear reviewer:
First of all, we would like to express our sincere appreciation to your valuable feedback. Your comments are highly insightful and enable us to significantly improve our manuscript. The following pages are our point-by-point responses to each of your comments.

[Figure]

**1 Response to general comments**

This paper presents a new solver and preconditioner for the barotropic mode of the POP ocean model, focusing on parallel performance. The new CSI solver is slower than the previous CG solver in serial, but it removes a parallel reduce operation which makes it significantly faster at high number of processes. The new EVP preconditioner reduces the number of iterations required by the solver, thus reducing both computation and communication time. The authors demonstrate that the new P-CSI solver significantly improves the efficiency of the POP model in massively parallel runs. The paper extends an earlier proceedings paper (Hu et al. 2015) by presenting: a more detailed description of the barotropic solver used in POP, analysis of the eigenvalues and condition number of the associated linear operator, analysis of the convergence rates of different solvers, and new estimates of the computational complexity of the solvers. It is however debatable whether the additional material merits another publication as most of the material originates from Hu et al. (2015). In addition the analysis of the condition numbers could be improved.

[Response]:

Yes. This paper extends our previous work originally presented in the 27th International Conference for High Performance Computing, Networking, Storage and Analysis (SC 2015) as we indicated in Section I. Most of the audiences in the SC conference are supercomputing specialists. Therefore, we simplified the background of the ocean model and focused on the design of algorithm, scalability tests and efficiency in the SC paper. However, we also hope that our work can be widely understood and accepted by climate modelers so that we expand the paper and submit to GMD by a completed new revision. This includes a large change in the review of barotropic mode and the current solver which may cause severe bottleneck in the large-scale computing; a new theoretical analysis in section 4 (provide a robust base for the approach) and a different view of the new results in section 5. Although some figures look similar, the stories behind these figures are totally different from the SC paper if you closely compare

them. For the major changes, please see our detailed reply to the reviewer 1. We will clearly introduce these differences in the revised manuscript if we have the opportunity.

**2   Response to specific comments**

(1) Section 4.1 Spectrum and condition number
As we are dealing with a 2D shallow water solver, this analysis would be clearer if it incorporated the non-dimensional barotropic CFL number. It would be useful to know what the CFL number of the 2D mode is in typical CESM runs, and use that as a basis for the analysis and idealized experiments.
Noting that $CFL = c\Delta t/\Delta x$, with $c = \sqrt{gH}$ being the speed of the gravity waves, $\phi$ in eq. (14)(and consequently the condition number) can be expressed as a function of CFL. Specifically $\phi = 1/(CFL_x CFL_y)$ and (using $\lambda_{min}, \lambda_{max}$ from line 273) the condition number is approximately $k = 4CFL^2 + 1$ for large time steps and aspect ratio =1, which shows the dependency clearly.
Similarly in Figs 5, 6 and 7 it would be useful to use the CFL number instead of the time step or 2D velocity. The value of the time step alone, for example, is not informative as it depends on how the idealized run was set up.
[Response]:
Thanks for your good suggestions. We will use the CFL numbers in Figs 5, 6 and 7 rather than time step sizes and 2D velocities in our revised manuscript. The non-dimensional barotropic CFL number will be used as the x-axis in Fig. 5 and the legends in Fig. 6 and Fig. 7.

(2) Section 3.2 A block EVP preconditioner
In the last paragraph the authors mention that the drawback of the EVP preconditioner is that it cannot be used to solve large problems due to propagation of errors. Does

this imply that EVP cannot be used at low processor counts? If so have the authors experimented with or documented the failure of EVP? Is it possible to derive a threshold problem size under which the EVP preconditioner is reliable?
[Response]:
Sorry for the confusing. This problem only refers to the original EVP approach (described in Roache, 1995). The standard EVP solver is already modified so that the method can be used for any domain size and any processor count using domain decomposition (e.g., Dietrich, 1975) regardless of parallelization. So it has no problem at all for any domain size. However, if the domain size is too large without using domain decomposition, the computation will be very slow (See the complexity analysis 4.3 when p=1). Using parallel domain decomposition can actually help and speed up the EVP solver. We will revise the sentence in line 230 to avoid the confusing and add the following two reference papers [3][4].
D. Dietrich (1975) 'Optimized Block-Implicit Relaxation ', Journal of Computational Physics, Vol.18, No.4 421-439.
Roache, P. J. (1995) Elliptic marching methods and domain decomposition, vol. 5, CRC press, 1995.

(3) Section 4.3 Computational complexity
These estimates of computational complexity are similar to those presented in Hu et al. (2015). Some of them are different however (e.g. eqns. 26, 27, 28) especially in terms of the computation time $T_c$, Why the difference?
[Response]:
As you pointed out, there is a minor difference in the new manuscript. In order to make our analysis more general and understandable, we remove the assumption of equal partition along longitude and latitude directions. Therefore, the meaning of $N$ is changed from partition number in both directions to the total grid number. The $O(\cdot)$ notation is also changed for the consistency. We also correct a minor error in which the computation time in P-CSI (with diagonal preconditioning) should be $T_c = 12N^2/P$

instead of $T_c = 13N^2/P$ to be precise.

(4) line 400: Fig. 8 shows that the P-CSI solver converges slower compared to PCG. Why is it so? The analysis in Section 4.2 concluded that the convergence rate should be similar.

[Response]:

We illustrated that the P-CSI has the same theoretical lower bound of convergence rate as PCG and ChronGear at page 11, line 324 when the estimation of eigenvalues is appropriate ($k' = k$). In practice, as we also illustrated at page 13, line 400, P-CSI usually converge slower than PCG with the same preconditioning. The reason is that P-CSI requires that $0 < \upsilon \le \lambda_i \le \mu(i = 1, 2, ..., N)$, which means that $k' = \mu/\upsilon \ge \lambda_{max}/\lambda_{min} = k$. According to Eq. (18) and Eq. (24), P-CSI has a slower convergence rate than PCG unless the estimation of eigenvalues is optimal. Furthermore, the theoretical bound is often too conservative for PCG. In practice, an increase in the convergence rate is often observed as the problem size increases, which is known as superlinear convergence of the PCG method [2]. To clarify, this explanation will be added into our revised version.

(5) line 415: It is not clear how the grid is divided in blocks. It might be worth explaining this is Section 2.1.

[Response]:

Thanks for your suggestions. POP divides the horizontal domain into small blocks evenly and distributes them to processes. We assume that there are N and M grids along longitude and latitude respectively, and the global domain is divided into n*m small blocks with size of N/n*M/m. These blocks are distributed to processors using simple Cartesian strategy or Space-filling Curve method [1].

We will add these materials into Section 2.1 to make the grid partition clear in our revised version.

**3 Response to technical corrections**

(1)line 109: $\rho_0$ is the constant reference density, not the actual water density.
[Response]: Thanks for your corrections. We will modify this error in our revised version.

(2) line 122: Meaning of the last sentence is unclear, please elaborate/reformulate.
[Response]: Thanks for your corrections. We will rewrite this sentence in our revised manuscript as follows.
"When we directly integrate the continuity equation from the bottom to the surface, we will get a form $\int_{-H}^{\eta} dz(\bigtriangledown \cdot u + \partial w/\partial z) = \partial \eta/\partial t + \bigtriangledown \cdot (H+\eta)U - q_w = 0$ under the surface boundary condition $w(\eta) = d\eta/dt - q_w = \partial \eta/\partial t + u(\eta) \cdot \bigtriangledown \eta - q_w$. The term including $\eta$ inside the divergence leads to a nonlinear elliptic system, so many mature numerical methods such as conjugate gradient method cannot handle this problem. To avoid this, POP linearizes the barotropic continuity equation by modifying the boundary condition from $w(\eta) = d\eta/dt - q_w = \partial \eta/\partial t + u(\eta) \cdot \bigtriangledown \eta - q_w$ to $w(\eta) = \partial \eta/\partial t - q_w$ and dropping a small term. More details can refer to [1]."

(3) line 170: typo: scalar
[Response]: Thanks for your corrections. We will correct "scaler" with "scalar" in our revised manuscript.

(4) line 285: Here the authors assume that the time step satisfies the CFL condition, i.e. CFL$\leq$1, where the velocity v is chosen rather arbitrarily. It would be better to use CFL numbers typical to CESM applications.
[Response]: Thanks for your corrections. To avoid the influence of experiment setup, we will use three non-dimensional barotropic CFL numbers (0.01, 1 and 5) as the legend of Fig. 7. The CFL number in realistic 0.1 degree run of POP is about 3.46, so we choose 0.01-5 as the range of CFL number.

(5) fig 7: what is the grid aspect ratio used in this test?

[Response]: The grid aspect ratio used here is 1, and we will add this information to make our manuscript rigorous.

(6) line 335: $T_c$ is not defined in this paper.

[Response]: The $T_c$ represents the computation complexity, we will add this definition in our revised manuscript.

(7) line 513: typo: scalar

[Response]: Thanks for your corrections. We will replace "scaler" with "scalar" in our revised manuscript.

(8) figs. 16 and 17: these figures are not mentioned in the manuscript

[Response]: Thanks for your corrections. We will remove these two figures in our revised version.

**4   References**

[1] Smith, R., et al. "The Parallel Ocean Program (POP) Reference Manual Ocean Component of the Community Climate System Model (CCSM) and Community Earth System Model (CESM)."ÂăRep. LAUR-01853Âă141 (2010).

[2] Beckermann, Bernhard, and Arno BJ Kuijlaars. "Superlinear convergence of conjugate gradients."ÂăSIAM Journal on Numerical AnalysisÂă39.1 (2001): 300-329.

[3] D. Dietrich (1975) 'Optimized Block-Implicit Relaxation ', Journal of Computational Physics, Vol.18, No.4 421-439.

[4] Roache, P. J. (1995) Elliptic marching methods and domain decomposition, vol. 5,

[Figure]

CRC press, 1995.

---

## Author Response (AR1)

We would like to thank the editor for the efforts in handling this manuscript, as well as the reviewers for their insightful and thoughtful reviews. These constructive comments further improve our manuscript. We have carefully addressed each comment and incorporated the changes in the revised manuscript accordingly. Our point-by-point responses are detailed as follows.

**1 Response to referee #1 (Dr. Muller)**

**1.1 Response to general comments**

In their paper: "P-CSI v1.0, an accelerated barotropic solver for the high resolution ocean model component in the Community Earth System Model v2.0" the authors discuss the implementation of a new preconditioned iterative solver algorithm for the solution of the elliptic PDE which arises in implicit time stepping of the barotropic mode in ocean models. By using an iterative method which avoids global communications, the scalability of the solver can be improved significantly on large core counts. The authors demonstrate that this leads to substantial performance improvements when running the model at high spatial resolution on more than 16,000 cores of the Yellowstone supercomputer.

Accurate and scalable models are absolutely essential for reliable predictions of the Earth's climate which have wide impact in the geoscience community. In the introduction the authors argue convincingly why the development of high resolution ocean models and of massively parallel elliptic solvers is necessary and their novel algorithm approach addresses an important bottleneck for scalability on large core counts (global parallel reductions). Since the barotropic solver accounts for a large fraction in the runtime, this work has a large impact for the numerical model they study and can also help to improve related models in atmospheric- and ocean-modelling. The work is put into context by referring to relevant related publications and the paper is very well written throughout, with the results supporting the theoretical analysis (in particular the theoretical performance analysis). The scientific results, in particular the use of a communication-avoiding iterative method as an alternative to "standard" Krylov subspace methods such as CG, are very interesting and the benefits of the method are demonstrated convincingly by detailed numerical experiments. As stated at the end of the introduction, this paper is based on a related conference proceedings publication [1], where the key ingredients of the algorithm and parallel scaling tests are described in detail. Compared to [1], the present GMD paper contains the following new material:

1.The properties of the discretized system and in particular the spectral radius of the matrix is derived for a simplified test setup (constant ocean depth). By a theoretical analysis of the convergence rate of the new P-CSI algorithm the authors argue that it converges as fast as the CG and ChronGear solvers which require additional global reductions.

2.Numerical estimates of the spectral radius for different aspect ratios and time step sizes are presented.

3.The detailed convergence history of different solvers/preconditioners is studied.

However, the key concepts of the solver/preconditioner setup and similar results are already given in [1] (for example for the convergence numbers, compare Fig. 6 in [1] and Figs. 9 11 in the present paper, Fig. 13-15 are a subset of results from [1] and as far as I can tell Fig. 10 and 12 are obtained with a similar setup as in [1]). I am therefore slightly concerned as to whether this paper contains sufficient new material to for a new publication, in particular since I'm not sure how relevant the variations in time step size really are in practice - the time step size is largely fixed by the CFL limit in other model components, such as the advective time scale (see further discussion below). To publish the paper it has to be made clear that large parts of it consist of new results. The solver code is made available online, but I was not able to compile and run it since it requires installation of the full model.

[Response]:

Thank you for your highly valued comments that our study has a large impact for the numerical model development and can help to improve related models in atmospheric and oceanic modelling.

This paper is an extended work originally presented in the 27th International Conference for High Performance Computing, Networking, Storage and Analysis (SC 2015) as we indicated in Section I. Most of the audiences in the SC conference are supercomputing specialists. Therefore, we simplified the background of the ocean model and focused on the design of algorithm, scalability tests and efficiency in the SC paper. We introduced our solver with some pseudo-code and used a lot of computer terminologies since the main readers are computer scientists. In order to expand the influence of our work to more general readers, we decide to extend our paper to GMD which is an outstanding academic exchange platform for climate modelers.

Therefore, we made a lot of changes to the content and structure here. We specifically summarized our major changes as follows and briefly described them in the introduction.

- After our presentation at SC conference in November 2015, many helpful advices were gathered. Some specialists suggested to provide more information about the universal applicability of our new solver in different cases/applications. Therefore, we theoretically analyzed the characteristics of P-CSI through the associated eigenvalues and their connection with the convergence rate here. The careful analysis also provided the main reasons why the proposed approach can lead to a significant improvement form many aspects, including the spectral radius of the matrix, condition number and the convergence rate. We showed that the P-CSI can converge as fast as the CG and ChronGear solvers which require additional global reductions. In SC paper, we only presented the computational complexity which is not completed.

- We provided more comprehensive reviews of barotropic mode and the associated solvers adopted in the original POP. We believe that this addition will help other climate modelers to comprehensively understand a general large-scale computing problem in POP, MOM, MITgcm, FVCOM, OPA models etc. The completed description helps the potential users to easily incorporate this approach to their own models. In SC paper, we only present a brief introduction of barotropic mode and the ChronGear solver.

- In order to make the climate modelers (instead of computer scientists) to better understand the new solver, we rewrote most of the sentences and moved all pseudo-codes and preconditioning procedure into the appendix for interested readers. We also avoided the use of obscure computer jargon to make it more readable. In addition, all figures have been redraw to emphasize the advantage of our proposed method and the overall performance of POP.

Section 5 presents some similar materials in the original manuscript and the SC paper because this section shows the same test case for the CESM POP. Thus, we listed the changes as follows:

(1) Fig. 8 verifies the theoretical analysis of the convergence rate of different barotropic solvers (not presented in the SC paper).

(2) Fig. 9 presents the individual timing for different phasees which combines some important information from the SC paper. This figure cannot be removed because it is very important to re-emphasize that the global reduction is the major bottleneck in the large-scale computing ($>O(10^3)$ processor counts) in the GMD paper. Also, the P-CSI with EVP preconditioner enhances the performance by significantly reducing the iteration number so that the time for

computation part is further reduced. We re-emphasize this major point but shorten the discussion.

(3) Fig. 10 shows the overall performance of CESM POP by including the timing for other components.. In the SC paper, we presented the timing analysis for barotropic solver only. This additional information is critical because the performance of 1/10 degree POP is not clearly documented yet. So, we put the total time (all components) to show the key advantage of our new approach.

About the unphysical time step size in Fig.5 (the new Fig. 4), we reanalyzed the solver property based on the realistic values (the first baroclinic wave speed of 2m/s and the gravity wave velocity of 200m/s). In the 0.1 degree ocean simulation, the CFL number is about $c\Delta t/\Delta x \approx 3.46$, thus the corresponding time step size are 17280s and 172.8s. We agree it makes more senses to use CFL number as the x-axis in some plots (see the responses below) instead of time-step (which is also connected by dx). Therefore, all relevant figures have been changed accordingly.

In conclusion, we believe this revised GMD manuscript will be an expanded, more completed, and more refined version of the conference paper. These results are further confirmed by a theoretical analysis. All figures have completely different stories (except the new Fig. 9 which reemphasizes the major bottleneck and how the new approach improves the timing) and key messages comparing the SC paper. We clearly introduced these differences in the revised manuscript.

**1.2 Response to specific comments**

(1) Early on in the introduction and in section 2.2 the authors mention that global reductions limit the performance of ChronGear and CG solvers, and this is one of the main motivations for using the P-CSI method. While this is clearly shown in Fig. 10, it might be good to already refer to numerical evidence here or quote numbers from [1]: which percentage of the runtime is spent in global reduction operations?

[Response]:

In the introduction, we added the following sentence into the introduction section ,

"For example, when approximately 4,000 cores are used, the global reduction in PCG and ChronGear takes approximately 74% and 68% of the whole barotropic time, respectively [1]. This situation will worsen with more cores." (Line 55 ∼ 58)

Also, we merged Fig. 10 with Fig. 12 into the new Fig. 9 so that we can reemphasize the advantage of our proposed method. The discussion is also shortened in section 5.

(2) In section 2 the authors derive the barotropic mode in the fundamental equations and then discretise it implicitly in time. At this point it might be good to briefly mention how this is related to other model components: how is the implicitly calculated height perturbation coupled back to the full equations? Are other parts of the equations (such as advection) solved explicitly? What are the typical time velocities/time scales(I assume that the implicit treatment of the barotropic mode is necessary since the gravity-wave velocity $\sqrt{g * H}$ is much larger than other velocities in the system, such as advective velocities - is this correct?).

[Response]:

Thanks for the suggestion. In the beginning of section 2.1, we added some brief introduction of the time scheme in POP. (Line $142 \sim 149$)

You are correct that the implicit approach is necessary to match the same time step between the barotropic and baroclinic modes. Hope the associated discussion has a more completed description.

(3) When showing strong scaling results such as in Figs. 3, 13-15 the number of unknowns per processor is relevant to assess the relative importance of halo exchange, could this information be added to the figures?

[Response]:

This advice is valuable and we added this information to Fig. 3 by including the new axis labeled as "number of grid points" . We eliminated Fig. 13 because it doesn't add too much. Figs. 14 and 15 are reduced into the new Fig. 10, which provides additional information about the overal timing.

(4) Discussion in section 4.1: the barotropic CFL number due to gravity waves of speed $c_g = \sqrt{gH}, n_g = c_g dt/dx$, is a very important quantity and for a given resolution directly related to the time step size. However, typically the time step size is limited by other processes in the model. For example, if there is another process with typical speed $c^{'}$, which is treated explicitly, then the related CFL number $n^{'} = c^{'} dt/dx$ is limited by $n^{'} < O(1)$. For example in atmospherical models $c_{advection} 10 \cdot c_{acoustic}$, and hence the $n_g$ should not be larger than $\approx$ 10. Could the authors include a discussion of this and also discuss physical limits on $n_g$ by referring to other components (i.e. non-barotropic) of the model? I think this is very important since the CFL number has a significant impact on the solver performance. By using $c_g = \sqrt{9.81 m/s^2 \times 4km} = 200 m/s$ the large time step sizes in Fig. 5 seems to be completely

unphysical if I assume that there is another explicitly treated process in the model which is $\approx 10\times$ slower than the gravity waves, but my intuition from atmospheric models might be misleading here and if the explicitly treated non-barotropic dynamics happens at much larger time scales then those large time steps make sense. This seems to be implied by the setup used for the 0.1 degree runs: assuming a depth of 4km, a time step size of 172.8s would lead to a CFL number of $\approx 10^4$. Since the condition number depends largely on the CFL number it would be good to see what the physically relevant values are.

[Response]:

Thanks for your corrections. Considering the limitation of CFL condition, the time step size in Fig. 5 is beyond the physical range indeed. The original purpose of this selection is to make it more intuitive to readers that the time step sizes have a large influence on the condition number of the coefficient matrix without taking the physical consistency into account. We redraw Fig.5 with the CFL number as the x-axis and make the values more credible in physics. In 0.1 degree ocean simulation, the time step size is $\Delta t$=172.8s (500 steps per simulation day), thus for the barotropic mode, the CFL number is about $c \cdot \Delta t/\Delta x \approx 3.46$ where $c = 200m/s$ is the gravity wave velocity in the barotropic mode and $\Delta x = 10000m$ is the horizontal grid length. If $c^{'} = 2m/s$ is a typical value for the first baroclinic wave speed, the CFL number is less than $c^{'} \cdot \Delta t/\Delta x \approx 0.035$. CFL numbers varying from 0.01 to 5 are used in Fig. 5 (the new Fig. 4) to cover more physically relevant cases in POP.

(5) The condition $dt = dx/v$, which is imposed at the bottom of page 9 should be clarified. The authors refer to $v$ as the "barotropic velocity" and then vary this between 2m/s and 200m/s. Should this v be some other velocity in the system which limits the time step size? The velocity relevant for the barotropic equation is the gravity wave speed $c_g = \sqrt{gH}$ which is 200m/s for a depth of 4km. Same question in section 5.1, where the authors fix $v = 2m/s$.

[Response]:

Changed. In the revised manuscript, we used the typical value for the first baroclinic wave speed 2m/s and the typical gravity wave velocity $c_g = \sqrt{gH} = 200m/s$ for a depth of 4km as the lower bound and the upper bound of the velocity range, respectively. Besides, we used the non-dimensional barotropic CFL number instead of velocity as the legend of Fig. 7 (the new Fig. 6) to show the dependency.

(6) It would help if the CFL number and (an estimate of the) condition number of the matrix are given for the realistic 0.1 degree run. Since the largest and smallest eigenvalue are estimated,

this information should be available.

[Response]:

We added this information in Section 4.1.

"In 0.1 degree realistic run, the CFL number is approximately $c \cdot \Delta t / \Delta x \approx 3.46$ (where $c = 200m/s$, $\Delta t = 172.8s$, and $\Delta x = 10000m$ are the typical gravity wave speed, time step and spatial resolution, respectively) and the condition number is approximately 250. Though the grid size of 0.1 degree POP is much larger than that of 1 degree POP, the condition number of 0.1 degree POP is smaller than the condition number of 1 degree POP (approximately 1200) owing to a smaller CFL number because of the small time step." (Line $334 \sim 339$)

(7) If the CFL numbers are very large (see previous points), then I really think that advanced preconditioners have the potential for improving the performance. Multigrid preconditioners could reduce the iteration count from O(100) to O(10), so might pay off even if one preconditioner application is more expensive.

[Response]:

As you point out, any advanced preconditioner which can quickly reduce the iteration count will be very useful to improve the performance. In fact, the EVP solver is a direct fast solver so that it has this capability and is the main reason we choose here. Furthermore, as we indicated in lines 66-77 of section 1, the multigrid method is a well-known scalable and efficient approach to solve the elliptic systems too. However, some related works confirmed that the geometric multigrid in global ocean models does not always scale ideally because of the presence of complex topography (land particularly), non-uniform or anisotropic grids ([2], [3], [4], [5], [6]). These constraints lead to an elliptic system with variable coefficients defined on an irregular domain in POP and complicate the modeling system. The algebraic multigrid (AMG) is an alternative to the geometric multigrid to handle the complex topography. However, the AMG setup in the parallel environment is more expensive than the iterative solver in climate modelling, which makes it unfavorable as a preconditioner [2]. On the contrary, the EVP preconditioner is simple enough and can effectively reduce the condition number of coefficient matrix by about 5 times in both 1 and 0.1 degree cases, which leads to a reduction of 2/3 iterations. Therefore, we use the direct EVP solver.

In the revised manuscript, we added the above discussions in the end of section 5.1. (Line $461 \sim 466$)

(8) In Fig. 5 it would be good to indicate the range of typical physical time step sizes for

each resolution instead of just plotting a wide range of time scales.

[Response]:

We replaced the time step sizes in the x-axis of Fig. 5 (the new Fig. 4) with the non-dimensional CFL number and make the values more credible in physics.

(9) Page 13, line 394: while the matrix becomes more ill conditioned as the problem size increases, the condition $dt = dx/v$ will limit this growth, in fact the upper bound on the condition number is at the order of $\approx gH/v^2$.

[Response]:

We added the above information which is confirmed by Fig. 7 (the new Fig. 6). (Line $443 \sim 444$)

(10) The theoretical analysis is carried out for a constant ocean depth H. How reasonable is this assumption and which impact do variations in H have?

[Response]:

The purpose of this assumption is to simply demonstrate the properties of the sparse matrix used in the POP. It is not general but provides a bound for cases with various H. An analysis for a variable ocean depth is now presented in our revised manuscript.

**1.3 Response to typos/minor comments**

(1) at several places in the paper "scaler" should be replaced by "scalar"

[Response]:

Corrected.

(2) to me "boundary communication" is a slightly unusual expression, I'd call this "halo exchange" since "boundary" could refer to a physical boundary in the global domain (such as the ocean-land interface).

[Response]:

Changed.

(3) at the bottom of page 4: should this read "[...] the barotropic continuity Eq. (4) *has been* linearised [...]" ("is linearised" implies that another term has to be removed from (4) to obtain a linear equation, but (4) is already linear).

[Response]:

We changed "is linearized" into "has been linearized" in the revised manuscript and added a

brief introduction about the linearization procedure in section 2. (Line 157 ∼ 164)

(4) bottom of page 8: "spectrum radius" -> "spectral radius"

[Response]:

Changed.

(5) definition of $P_k(\xi)$ between Eqs. (19) and (20) on page 10: What are $\alpha$ and $\beta$ here?

[Response]:

We added the definition of $\alpha$ and $\beta$ in our revised version. The $\alpha = \frac{2}{\mu - \nu}$ and $\beta = \frac{\mu + \nu}{\mu - \nu}$. (Line 360)

(6) in appendix A and B it might help if the global reduction operations in steps 2. and 5. of PCG and steps 3. and 4. of ChronGear are highlighted. Also, a sentence to the appendix which clarifies that the global reduction of $\rho_k$ and $\sigma_k$ in the ChronGear algorithm can be combined (thus halving the latency) might help.

[Response]:

We highlighted these steps with rectangle blocks and added some sentences to clarify the global reduction operations in appendix A. (Line 562, 565, 573, 583, 584, 594 ∼ 595)

(7) Fig. 4 does not add relevant information and should be removed.

[Response]:

We removed Fig.4 and related paragraphs in our revised version.

(8) Fig. 2: replace "sparse pattern" -> "sparsity pattern"

[Response]:

Changed.

**2 Response to anonymous referee #2**

**2.1 Response to general comments**

This paper presents a new solver and preconditioner for the barotropic mode of the POP ocean model, focusing on parallel performance. The new CSI solver is slower than the previous CG solver in serial, but it removes a parallel reduce operation which makes it significantly faster at high number of processes. The new EVP preconditioner reduces the number of iterations

required by the solver, thus reducing both computation and communication time. The authors demonstrate that the new P-CSI solver significantly improves the efficiency of the POP model in massively parallel runs. The paper extends an earlier proceedings paper (Hu et al. 2015) by presenting: a more detailed description of the barotropic solver used in POP, analysis of the eigenvalues and condition number of the associated linear operator, analysis of the convergence rates of different solvers, and new estimates of the computational complexity of the solvers. It is however debatable whether the additional material merits another publication as most of the material originates from Hu et al. (2015). In addition the analysis of the condition numbers could be improved.

[Response]:

Yes. This paper extends our previous work originally presented in the 27th International Conference for High Performance Computing, Networking, Storage and Analysis (SC 2015) as we indicated in Section I. Most of the audiences in the SC conference are supercomputing specialists. Therefore, we simplified the background of the ocean model and focused on the design of algorithm, scalability tests and efficiency in the SC paper. However, we also hope that our work can be widely understood and accepted by climate modelers so that we expand the paper and submit to GMD by a completed new revision. This includes a large change in the review of barotropic mode and the current solver which may cause severe bottleneck in the large-scale computing; a new theoretical analysis in section 4 (provide a robust base for the approach) and a different view of the new results in section 5. Although some figures look similar, the stories behind these figures are totally different from the SC paper if you closely compare them. For the major changes, please see our detailed reply to the reviewer 1. These main differences are summarized in the revised section 1.

**2.2 Response to specific comments**

(1) Section 4.1 Spectrum and condition number

As we are dealing with a 2D shallow water solver, this analysis would be clearer if it incorporated the non-dimensional barotropic CFL number. It would be useful to know what the CFL number of the 2D mode is in typical CESM runs, and use that as a basis for the analysis and idealized experiments.

Noting that $CFL = c\Delta t/\Delta x$, with $c = \sqrt{gH}$ being the speed of the gravity waves, $\phi$ in eq. (14)(and consequently the condition number) can be expressed as a function of CFL. Specifically $\phi = 1/(CFL_x CFL_y)$ and (using $\lambda_{min}, \lambda_{max}$ from line 273) the condition number is approximately $k = 4CFL^2 + 1$ for large time steps and aspect ratio =1, which shows the dependency clearly.

Similarly in Figs 5, 6 and 7 it would be useful to use the CFL number instead of the time step or 2D velocity. The value of the time step alone, for example, is not informative as it depends on how the idealized run was set up.

[Response]:

Thanks for the suggestions. We used the CFL numbers in Figs 5, 6 and 7 (our new Figs 4, 5, 6)rather than time step sizes and 2D velocities in our revised manuscript. The non-dimensional barotropic CFL number is used as the x-axis in Fig. 5 (new Fig. 4) and the legends in Fig. 6 (new Fig. 5) and Fig. 7 (new Fig. 6).

(2) Section 3.2 A block EVP preconditioner

In the last paragraph the authors mention that the drawback of the EVP preconditioner is that it cannot be used to solve large problems due to propagation of errors. Does this imply that EVP cannot be used at low processor counts? If so have the authors experimented with or documented the failure of EVP? Is it possible to derive a threshold problem size under which the EVP preconditioner is reliable?

[Response]:

Sorry for the confusing. This problem only refers to the original EVP approach (described in Roache, 1995). The standard EVP solver is already modified so that the method can be used for any domain size and any processor count using domain decomposition (e.g., Dietrich, 1975) regardless of parallelization. So it has no problem at all for any domain size. However, if the domain size is too large without using domain decomposition, the computation will be very slow (See the complexity analysis in section 4.3 when p=1). Using parallel domain decomposition can actually help and speed up the EVP solver.

Therefore we revised the corresponding sentences to avoid the confusing. (Line 280 ∼ 284)

D. Dietrich (1975). Optimized Block-Implicit Relaxation, Journal of Computational Physics, Vol.18, No.4 421-439.

Roache, P. J. (1995). Elliptic marching methods and domain decomposition, vol. 5, CRC press, 1995.

(3) Section 4.3 Computational complexity

These estimates of computational complexity are similar to those presented in Hu et al. (2015). Some of them are different however (e.g. eqns. 26, 27, 28) especially in terms of the computation time $T_c$, Why the difference?

[Response]:

As you pointed out, there is a minor difference in the new manuscript. In order to make our analysis more general and understandable, we remove the assumption of equal partition along longitude and latitude directions. Therefore, the meaning of $\mathcal{N}$ is changed from partition number in both directions to the total grid number. The $O(\cdot)$ notation is also changed for the consistency. We also correct a minor error in which the computation time in P-CSI (with diagonal preconditioning) should be $T_c = 12\mathcal{N}^2/P$ instead of $T_c = 13\mathcal{N}^2/P$ to be precise.

(4) line 400: Fig. 8 shows that the P-CSI solver converges slower compared to PCG. Why is it so? The analysis in Section 4.2 concluded that the convergence rate should be similar.

[Response]:

We illustrated that the P-CSI has the same theoretical lower bound of convergence rate as PCG and ChronGear at page 11, line 324 when the estimation of eigenvalues is appropriate $(k' = k)$. In practice, as we also illustrated at page 13, line 400, P-CSI usually converge slower than PCG with the same preconditioning. The reason is that P-CSI requires that $0 < \upsilon \leq \lambda_i \leq \mu (i = 1, 2, ..., N)$, which means that $k' = \mu/\upsilon \geq \lambda_{max}/\lambda_{min} = k$. According to Eq. (18) and Eq. (24), P-CSI has a slower convergence rate than PCG unless the estimation of eigenvalues is optimal. Furthermore, the theoretical bound is often too conservative for PCG. In practice, an increase in the convergence rate is often observed as the problem size increases, which is known as superlinear convergence of the PCG method [2]. To clarify, this explanation is added into our revised version. (Line $449 \sim 457$)

(5) line 415: It is not clear how the grid is divided in blocks. It might be worth explaining this is Section 2.1.

[Response]:

Thanks for your suggestions. We added additional discussion about the grid partition in the new Section 2.1. (Line $206 \sim 209$)

**2.3 Response to technical corrections**

(1)line 109: $\rho_0$ is the constant reference density, not the actual water density.

[Response]:

Corrected.

(2) line 122: Meaning of the last sentence is unclear, please elaborate/reformulate.

[Response]:

This sentence is revised as follows.

"When we directly integrate the continuity equation from the bottom to the surface, we obtain a form $\int_{-H}^{\eta} dz(\nabla \cdot \mathbf{u} + \frac{\partial w}{\partial z}) = \frac{\partial \eta}{\partial t} + \nabla \cdot (H + \eta)\mathbf{U} - q_w = 0$ under the surface boundary condition $w(\eta) = \frac{d\eta}{dt} - q_w = \frac{\partial \eta}{\partial t} + \mathbf{u}(\eta) \cdot \nabla\eta - q_w$. The term including $\eta$ inside the divergence leads to a nonlinear elliptic system; thus, many mature numerical methods such as the conjugate gradient method cannot handle this problem. To avoid this, the POP linearizes the continuity equation by dropping the divergence term in the boundary condition, which becomes $w(\eta) = \frac{\partial \eta}{\partial t} - q_w$. Equation (4) is the barotropic continuity equation which has been linearized." (Line 157 $\sim$ 164)

(3) line 170: typo: scalar

[Response]:

Corrected.

(4) line 285: Here the authors assume that the time step satisfies the CFL condition, i.e. CFL$\leq$1, where the velocity v is chosen rather arbitrarily. It would be better to use CFL numbers typical to CESM applications.

[Response]:

To avoid the impact of experiment setup, we used five non-dimensional barotropic CFL numbers (0.01, 0.1, 0.5, 1 and 5) as the legend of Fig. 7 (the new Fig. 6). The CFL number in realistic 0.1 degree run of POP is about 3.46, so we choose 0.01-5 as the range of CFL number.

(5) fig 7: what is the grid aspect ratio used in this test?

[Response]:

The grid aspect ratio used here is 1 (the new Fig. 6), and this information is included now. (Line 331)

(6) line 335: $T_c$ is not defined in this paper.

[Response]:

The $T_c$ represents the computation complexity, we added this definition in our revised manuscript. (Line 391)

(7) line 513: typo: scalar

[Response]:

Corrected.

(8) figs. 16 and 17: these figures are not mentioned in the manuscript

[Response]:

We removed these two figures in our revised version.

**3 References**

[revised manuscript text omitted]

---

## Author Response (AR2)

We would like to thank the editor for the efforts in handling this manuscript, as well as the reviewers for their insightful and thoughtful reviews. These constructive comments further improve our manuscript. We have carefully addressed each comment and incorporated the changes in the revised manuscript accordingly. Our point-by-point responses are detailed as follows.

**1 Response to referee #1 (Dr. Muller)**

**1.1 Response to general comments**

I think the document has been improved significantly and the authors argue convincingly as to how their work extends what is presented in the SC proceedings article. I would like to thank the authors for addressing the comments in the reviews. Below are a few further (minor) comments which I suggest taking into account before publication. In particular I would still suggest clarifying the definition and discussion of the CFL number in section 4.1 (see also comments on original submission by reviewer 2).

(1) The overview of changes with respect to the SC paper in lines 104 - 124 is good, but I'd suggest shortening this a bit. Comments like "we rewrote most of the sentences" (line 121) or "After our presentation at the SC conference in November 2015, much helpful advice was gathered" (lines 104 - 105) don't really add any useful information and I don't think the reader will be interested in all the details as to why and how exactly the material was reorganised.

[Response]:

Thanks, we have shortened this part in our revised manuscript to make this contribution more concise while emphasizing the significant changes.

(2) last sentence in caption of Fig. 10: "can refer to Fig. 3" -> "can be deduced from Fig. 3" (or something similar)

[Response]:

Corrected.

(3) In general I think some of the new material would benefit from polishing the English to make sure it is of the same standard as the rest of the article.

[Response]:

Thank you very much for your comment. We reiterated the manuscript once again among authors and further polish the English.

(4) I think it is good that some of the Figures have been removed/merged

[Response]:

Thanks.

(5) Fig 3: It should be made clear that this figure refers to the number of unknowns per processor, so I would replace "Number of Grid points" -> "Number of unknowns per processor". The rule of thumb is that strong scaling breaks down once you have $\approx 1000 - 5000$ unknowns per processor, so on the right you are deep in the strong scaling limit. This is consistent with Fig. 9: the halo exchange time is of the same size or even larger than the computation time for $> 2000$ cores.

[Response]:

We have modified this in Fig. 3.

(6) In section 4 $\Delta t$ is used for the time step size, but in section 2.1 it is $\tau$. Could the same notation be used throughout the paper to avoid confusion?

[Response]:

Corrected.

(7) I think the discussion of the CFL number in section 4.1 is clearer now, but there are still bits that are very confusing, in particular the exact definition of the CFL number. In line 311 the CFL number in introduced as $CFL = v\Delta t/\Delta x$, which seems to imply that $v$ is some other velocity (i.e. *not* the barotropic velocity $\sqrt{gH}$, which appears separately in the definite of $\Phi$ in line 310. However, then at the end of the section (line 335) the CFL number is redefined as $CFL = c\Delta t/\Delta x$, which is the CFL number of gravity waves. The CFL number of explicitly treated modes is $\approx 100$ smaller than 3.46 (otherwise they would not be stable). Can you therefore clarify that $CFL = v * \Delta t/\Delta x$ is the CFL number for the *non-barotropic* modes in the system, i.e. modes which propagate with $v \ll \sqrt{gH}$? I understand that in this section you concentrate on variation in the CFL number (due to variations in time step size, grid resolution and $v$) and their impact on the solver performance, but still think that a discussion of the relative size of $v$ and $\sqrt{gH}$ would be useful here. E.g. saying something like: "$\sqrt{gH} \sim 200m/s$ is the speed of the gravity waves which are treated implicitly in the barotropic solver. $v$ is the speed of other, explicitly treated processes in the system and typically $v < \sqrt{gH}$. While in the POP model at 0.1 resolution $v \sim 2m/s$ and $CFL \approx 0.0346$, in this section we also consider

a set of velocitities $v$ and a wider range of CFL numbers to make more general statements on the algorithmic performance of the solver." Alternatively, to avoid the confusion above, the (barotropic) CFL number could be defined as $CFL = c\Delta t/\Delta x = 3.46$ throughout the section. Then simply $\Phi = 1/CFL^2$, as also noted by reviewer 2.

[Response]:

We agree with your justification due to too many degrees of freedom. We have removed the discussions of v = 2 m/s and v = 20 m/s, and we have also set $v = \sqrt{gH}$ to make the form of $\Phi$ and related analysis more clear.

(8) top of page 18: typo, "extrem" -> "extreme"

[Response]:

Corrected.

(9) line 158/158: equation is broken very awkwardly at the ·

[Response]:

Modified.

(10) How is $\overline{H}$ defined in appendix A? If it is the average depth, and $\min H \leq \overline{H} \leq \max H$, then not both inequalities in (A3) and (A4) can be true. The last inequality in (A4) should be $\geq 2\min(\alpha - 1/\alpha, 1/\alpha - \alpha)\min H + \Phi \max H$. Eq. (17) would have to be adjusted accordingly.

[Response]:

Yes, the $\overline{H}$ in appendix A is the average depth, and we have added a sentence "where $\overline{H}$ is defined in Section 2.1" at line 532 to illustrate the definition of $\overline{H}$, we think that the inequality in (A4) is true because $2\min(\alpha - 1/\alpha, 1/\alpha - \alpha)$ is less than or equal to zero, thus, $2\min(\alpha - 1/\alpha, 1/\alpha - \alpha)\overline{H} \geq 2\min(\alpha - 1/\alpha, 1/\alpha - \alpha)\max(H)$.

**2 Response to anonymous referee #2**

**2.1 Response to general comments**

The manuscript has been improved since the first version. The following aspects however should be addressed before publication.

(1) Section 2.1: For better readability, present the barotropic system (3)-(4) before going into the details of the barotropic mode (The implicit treatment of ...) I'd also introduce the CFL

number already here as it is a property of the numerical system: The rationale why the 2D mode is treated implicitly is really the strict CFL condition.

[Response]:

We have adjusted the sequence of sentences in the range between line 124 and line 149, and we have also introduced the barotropic CFL number as $CFL = \frac{\sqrt{gH} \cdot \tau}{\Delta x}$ in line 143.

(2) line 310: The authors write $Phi = (v^2)/(gHCFL^2)$. As we are dealing with a 2D shallow water solver, why the authors do not set v=sqrt(gH) and use $Phi = 1/CFL^2$ instead? That would greatly simplify the subsequent analysis, as you'd only need to vary the CFL number. In fig 4 for instance, varying both v and the CFL number does not seem meaningful. On line 334 the authors themselves state that the used velocity scale is the surface gravity wave speed, hence sqrt(gH).

[Response]:

Thank you very much for the suggestion. We have set $v = \sqrt{gH}$ to make the analysis more clear and introduced the barotropic CFL number in Section 2.1.

**2.2 Response to technical corrections**

(1) line 112: Please review for clarity: "In the SC paper, we only presented the computational complexity which is not completed."

[Response]:

We have removed this sentence in our revised manuscript.

(2) line 266: typo: improves improve

[Response]:

Corrected.

(3) line 319: poor grammar: When the aspect ratio of the horizontal grid cell *approaches* unity

[Response]:

We have modified this poor grammar in line 300, line 303 and line 308.

(4) line 337: reformulate sentence for better readability

[Response]:

We have shortened the original sentence as "For comparison, the condition number in the 1 degree POP simulation is higher, which is approximately 1200".

(5) line 443: "...until it reaches the upper bound at the order of $gH/v^2$" Meaning unclear.

[Response]:

We have rewritten this sentence as "As shown in Fig. 7, when the problem size increases, the coefficient matrix becomes more poorly conditioned, thus increasing the number of iterations" to make its meaning more clear.

(6) line 512: duplicate sentence

[Response]:

Corrected.

(7) throughout the text: units should be printed with normal not italic typeface, separated with a space: e.g. 4 km

[Response]:

Corrected.

[revised manuscript text omitted]